# Cast shadows reveal changes in glacier surface elevation

Monika Pfau[1], Georg Veh[1], Wolfgang Schwanghart[1]

[1]Department of Environmental Sciences and Geography, University of Potsdam, Potsdam, 14476, Germany

*Correspondence to*: Monika Pfau (monika.pfau@arcor.de)

**Abstract.** Increased rates of glacier retreat and thinning need accurate local estimates of glacier elevation change to predict future changes in glacier runoff and their contribution to sea level rise. Glacier elevation change is typically derived from digital elevation models (DEMs) tied to surface change analysis from satellite imagery. Yet, the rugged topography in mountain regions can cast shadows onto glacier surfaces, making it difficult to detect local glacier elevation changes in remote areas. A rather untapped resource are precise, time-stamped metadata on solar position and angle in satellite images. These data are useful to simulate shadows from a given DEM. Accordingly, any differences in shadow length between simulated and mapped shadows in satellite images could indicate a change in glacier elevation relative to the acquisition date of the DEM. We tested this hypothesis at five selected glaciers with long-term monitoring programs. For each glacier, we projected cast shadows on the glacier surface from freely available DEMs and compared simulated shadows to cast shadows mapped from ~40 years of Landsat images. We validated the relative differences with geodetic measurements of glacier elevation change where these shadows occurred. We find that shadow-derived glacier elevation changes are consistent with independent photogrammetric and geodetic surveys in shaded areas. Accordingly, a shadow cast on Baltoro Glacier (Karakoram, Pakistan) suggests no changes in elevation between 1987 and 2020, while shadows on Great Aletsch Glacier (Switzerland) point to negative thinning rates of about 1 m per year in our sample. Our estimates of glacier elevation change are tied to occurrence of mountain shadows, and may help complement field campaigns in regions that are difficult to access. This information can be vital to quantify possibly varying elevation-dependent changes in the accumulation or ablation zone of a given glacier. Shadow-based retrieval of glacier elevation changes hinges on the precision of the DEM as the geometry of ridges and peaks constrain the shadow that we cast on the glacier surface. Future generations of DEMs with higher resolution and accuracy will improve our method, enriching the toolbox for tracking historical glacier mass balances from satellite and aerial images.

# 1 Introduction

Quantifying spatial and temporal patterns of glacial changes is important to understand the response of the cryosphere to ongoing atmospheric warming (IPCC 2019). Changes in glacier volume determine the availability of regional and local freshwater resources that support the basic needs of many millions of people living in glaciated river basins (IPCC 2019; Pritchard 2019; Azam et al. 2021). Glacier retreat can shift ecosystems higher in elevation, changing the composition of, and possibly creating new, habitats (Brighenti et al. 2019; Cauvy-Fraunié and Dangles 2019). Shrinking glaciers also alter discharge seasonality, enhance rates of sediment transport, and shift biogeochemical and contaminant fluxes in glaciated river basins (Milner et al. 2017; Li et al. 2021). In high mountains, glacier retreat can also destabilize adjacent hillslopes, possibly enhancing the frequency and magnitude of catastrophic slope failures (Huggel et al. 2012). Other hazards to mountain communities evolve from new meltwater lakes that can suddenly empty in glacial lake outburst floods (Veh et al. 2020). Recent appraisals entail that ice loss has accelerated globally in past decades, with thinning rates of glaciers outside the Antarctic and Greenland ice sheets having doubled between 2000 and 2019 (Hugonnet et al. 2021). Still, some 141,000 km³ of glacier ice cover ~10% of the Earth's land surface today (Farinotti et al. 2019; Millan et al. 2022). Given projected future warming scenarios, sustainable management of these remaining ice resources requires accurate knowledge of regional and local mass balances (Richardson and Reynolds 2000; Bolch et al. 2011).

Measuring changes in the surface elevation of glaciers relies on repeated field and remote-sensing based surveys. Space-borne techniques such as laser altimetry (e.g., ICESat) (Moholdt et al. 2010; Neckel et al. 2014), radar interferometry (Farías-Barahona et al. 2020) or stereo-photogrammetry (Bolch et al. 2011) helped quantify changes in glacier surface elevation over large spatial scales and in terrain which is difficult to access. These appraisals are largely constrained to the past two decades (Belart et al. 2020; Geyman et al. 2022; Mannerfelt et al. 2022), with few exceptions such as Corona and Hexagon missions, which provided one-time stereo image pairs between the 1960s and 1970s (Lovell et al. 2018; Dehecq et al. 2020). Other space-borne derived estimates of long-term glacier changes have relied on time series of optical satellite images, yet without the capability of using stereo-photogrammetry. The Landsat mission has been particularly useful for mapping changes in glacier area, rather than elevation, primarily due to continuous recording period extending back to the 1970s, the high temporal revisit rate of 16 days, and a moderate spatial resolution of 30 m in the visible to shortwave infrared electromagnetic spectrum (Paul et al. 2011; Wulder et al. 2019; Wulder et al. 2022). If intersected with a DEM, glacier outlines mapped from Landsat (or any other satellite or aerial) images can be used to estimate changes in glacier elevation (Rankl and Braun 2016; Zhang et al. 2016).

While optical satellite and aerial imagery provides the longest, remotely sensed records of glacier change, its analysis is challenging in topographic settings where high relief casts shadows on highly reflective glacier surfaces (Kääb et al. 2016). As mountains block the direct incoming solar radiation, shaded glacier surfaces are characterized by a low variation of radiometric values, thus complicating visual image interpretation or automated approaches of image classification (Richter

1998; Paul et al. 2002; Racoviteanu and Williams 2012; Li et al. 2016). The problem of cast shadows increases with latitude owing to seasonal differences in solar elevation angle, and with the height of mountains, as those can cast wider shadows. Against these known limitations, we hypothesize that cast shadows in optical satellite images also have a largely untapped

60   potential for mapping glacier elevation changes. If the local glacier elevation has changed in two successive time steps, the shape of shadows emanating from adjacent mountains has to change accordingly, as long as solar elevation, azimuth, and the geometry of ridges and peaks remain constant (Fig. 1). Therefore, we expect that glacier thinning must locally cause longer shadows, while a local gain in glacier thickness will shorten the length of shadows. Using the tangent, the horizontal offset can be converted into a vertical displacement, i.e. a change in elevation. These changes in elevation can also be translated into

65   estimates of glacier altitude using a DEM as a reference (Fig. 1).

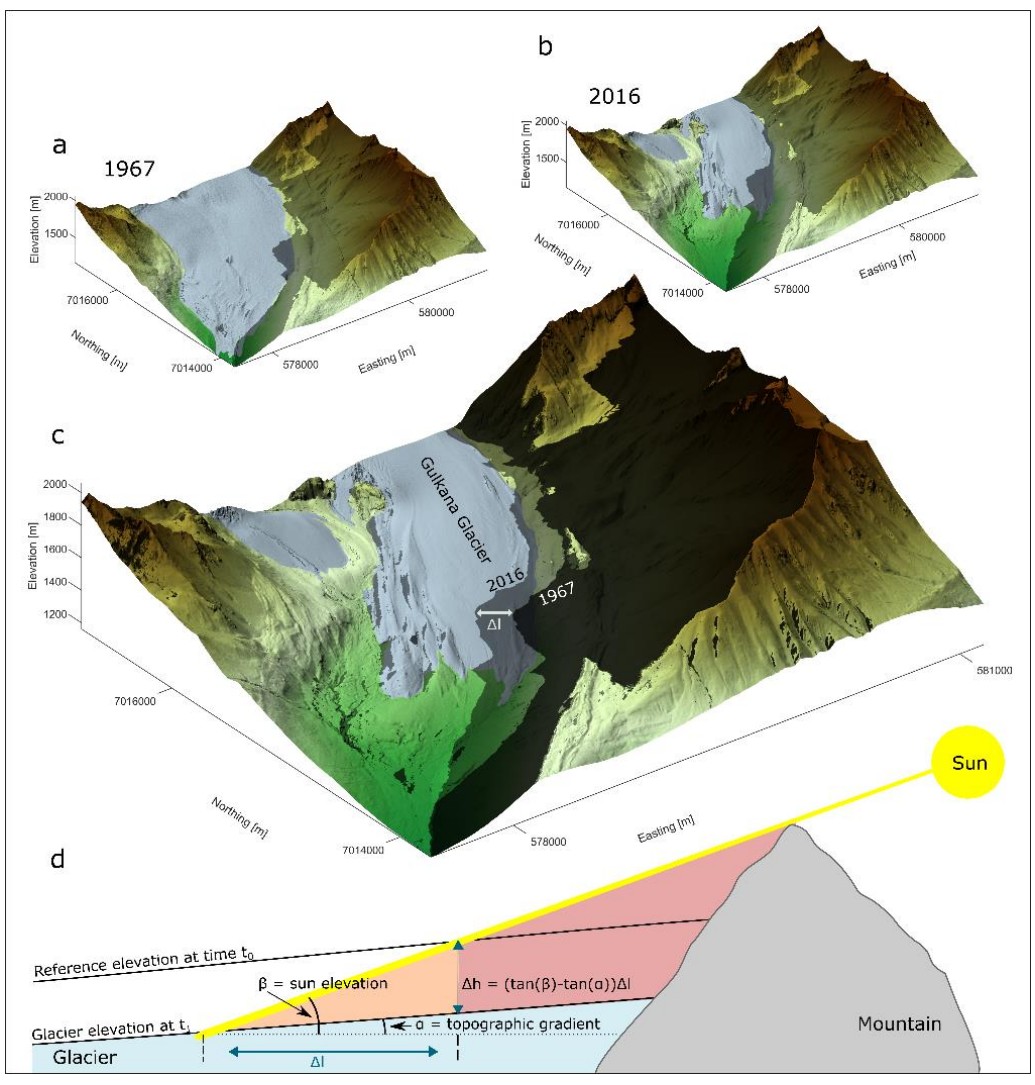

**Figure 1: Effects of changing glacier elevation on the length of cast shadows.** Example of modelled shadows on Gulkana Glacier, Alaska, using digital elevation models and mapped glacier outlines in two distinct years from McNeil et al (2022). **(a)** DEM from and surface area (light blue) of Gulkana Glacier in 1967. **(b)** DEM from and surface area of Gulkana Glacier in 2018. **(c)** DEM from 2018 with shadows from 1967 and 2018. Shadows were calculated based on a sun elevation of 20° and sun azimuth of 135°. The horizontal difference between the shadows (arrow in **c**) is 210 m. **(d)** Diagram of the trigonometric relationship that predicts longer horizontal shadows under a constant sun elevation β and mountain topography, assuming that the glacier maintains its topographic gradient α. In the example, the gain in shadow length at the terminus of the Gulkana Glacier translates into a glacier elevation change of ca. -76 m.

Few studies have explored the potential of cast shadows in satellite images to detect surface changes of glaciers. A recent study, for example, assessed ice-shelf freeboard heights of the Abbot ice-shelf, Antarctica (Rada Giacaman 2022). Another appraisal assessed the potential of the method for the Aletsch Glacier using Sentinel-2 for the period 2017-2021 (Dematteis et al. 2023). Yet, the potential of cast-shadows in glacier geodetic surveys has remained unaddressed on a broader geographic range and over longer time scales. Here we address the question of how well, or if at all, we can measure elevation changes on glaciers based on the variability of shadows cast by surrounding mountains. To this end, we develop and test an approach that applies trigonometry to time series of shadows extracted from Landsat satellite images from 1986 to 2021, draped over local DEMs, in order to identify local glacier surface changes. We validate this method at five glaciers for which we have detailed information on local glacier elevation changes.

## 2 Study sites

We selected glaciers in North America, Europe, and Central Asia, spanning 20° of latitude on the northern hemisphere (Fig. 2). Our selection was guided by the availability of decadal time series of glacier mass balances and high-resolution DEMs, and glacier outlines, providing a validation to our analysis. The shadows cast on these glaciers account for varying sun angles and surrounding relief, and occur in accumulation as well as ablation areas.

The Great Aletsch Glacier is located in the Swiss Alps, offering one of the longest consecutive records of mass balances in this mountain region (Bauder et al. 2007). The summit of Dreieckhorn casts a pronounced shadow on the Great Aletsch Firn at ~2,950 m a.s.l., which is close to the estimated equilibrium line altitude (ELA) of 2,961 m during the period of 1971-1990 (Zemp et al. 2007). High and steep mountains surround Baltoro Glacier in Pakistan. The Mitre Peak creates a nearly triangular shadow near Concordia (~4,500 m a.s.l.), which is the confluence of Baltoro and Godwin-Austen Glacier. This shadow is likely in the ablation zone, given an ELA at ~5,200 m a.s.l. (Minora et al. 2015). The northern-most glacier in our study is Gulkana Glacier (Alaska, USA), shaded by Ogive Mountain at ~1,850 m a.s.l. in the west and by Icefall Peak at ~1,800 m a.s.l. in the east. We did not study the shadow near the tongue of Gulkana Glacier, given that most Landsat images are acquired at noon when shadows are absent or very small. The ELA of Gulkana Glacier ranged from 1,811 m a.s.l. to 2,178 m a.s.l. between 2009 and 2019 (McNeil et al. 2022), so that the shadows were largely in the ablation zone. On South Cascade Glacier (Washington, USA), Lizard Mountain has two peaks, which form one coherent shadow on the glacier (~2,050 m a.s.l.). This shadow is above the ELA, which ranges between 1,794 and 2,042 m a.s.l. (1986 to 2018) (McNeil et

al. 2022). Finally, Sperry Glacier (Montana, USA) is shaded at an altitude of ~2,350 m a.s.l. by Gunsight Mountain. The shadow is situated largely in the ablation zone, given an average ELA at ~2,500 m a.s.l. for the period 2005-2019 (McNeil et al. 2022).

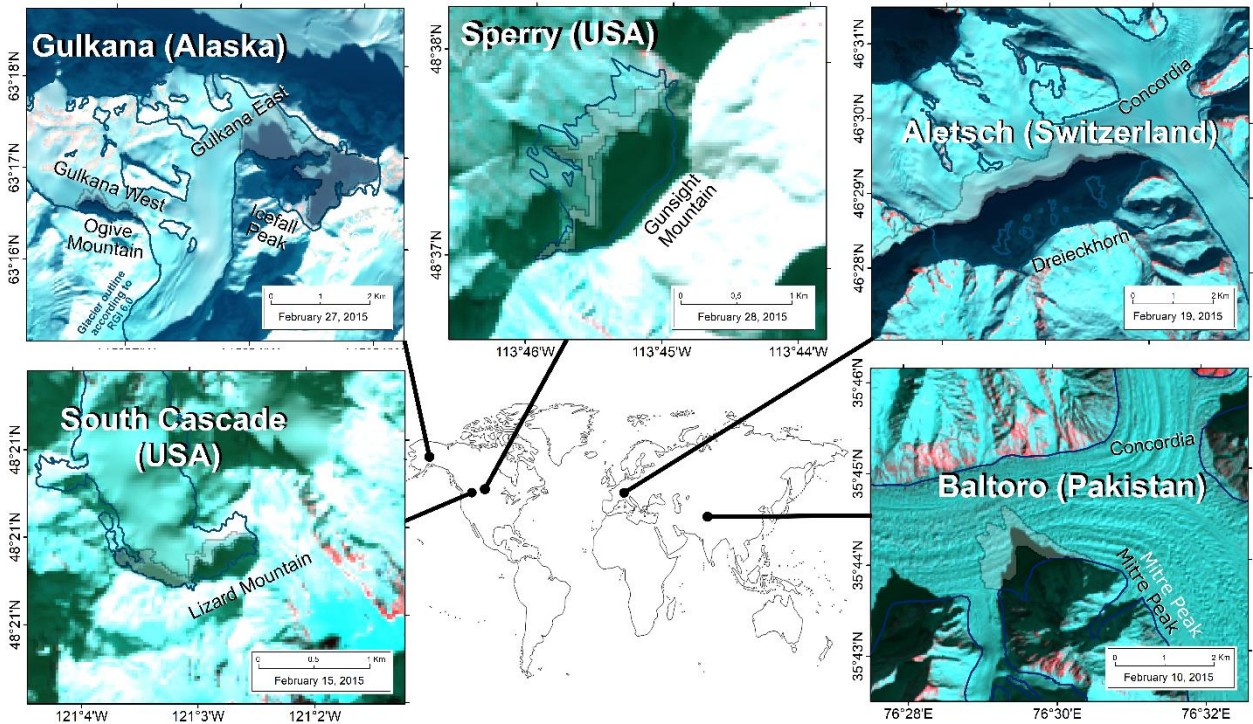

**Figure 2: Map of the five study regions.** Images are false-color composites (SWIR, blue, and green bands) from Landsat OLI obtained in
February 2015. Blue outlines are glaciers in the Randolph Glacier Inventory (RGI), V6.0. The semi-transparent areas show the difference between the largest and smallest shadow mapped in Landsat images in our study period, which we use for comparison with independent data and studies.

## 3 Data and methods

### 3.1. Satellite images and DEMs

We obtained 30-m resolution Landsat images (level L1TP-Precision and Terrain corrected) to map shadows within the glacier surface. To this end, we downloaded 69 cloud-free Landsat images (45 from TM, two from ETM+, and 22 from OLI) with acquisition period between 1986 and 2021 from the USGS *EarthExplorer* (https://earthexplorer.usgs.gov/, Appendix A1). L1TP images offer high radiometric and geodetic accuracy by using ground control points and correcting for topographic displacement using regional DEMs (https://www.usgs.gov/landsat-missions/landsat-levels-processing#L1TP).
We could not find any notable offsets between successive images in the time series.

We used several DEMs to simulate cast shadows for the dates at which the Landsat images were acquired (Table A2). For four glaciers, we used the DEM of the Shuttle Radar Topography Mission (SRTM-1, 1 arc second spatial resolution), which corresponds to a spatial resolution of 30 m in the local projection (Farr et al. 2007). Gulkana Glacier is located beyond the maximum acquisition range of SRTM at 60° North. We therefore used a 2-m stereo-photogrammetric DEM of

WorldView-1 data acquired in 2009, which is also part of the ArcticDEM (Porter et al. 2022). Owing to high vertical uncertainties in SRTM data in rough topography (Mukul et al. 2017; Liu et al. 2019), we used additional DEMs to enhance and validate our results. For Great Aletsch Glacier, we obtained the swissALTI3D DEM (acquisition year 2017-2018, version 2019, downsampled to 5 m spatial resolution by merging multiple raster datasets). For Baltoro Glacier, we replaced Mitre Peak (the source of the shadow cast on the glacier) in the SRTM-1 DEM using data from the Viewfinder Panoramas (VFP) project

(De Ferranti 2015). VPF is an improved version of the SRTM DEM drawing on auxiliary DEMs at locations where SRTM features voids or artefacts due to phase unwrapping errors. In the Higher Himalayas, the accuracy of the SRTM DEM decreases as elevation and steepness increase (Mukul et al. 2017, Liu et al. 2019). Indeed, the original SRTM-3 DEM (3 arc seconds or approximately 90 m) features a void at Mitre Peak, and suggests that its elevation was interpolated (EROS 2018). We therefore filled this void using the VFP DEM while maintaining the elevation of the glacier from the original SRTM DEM. We also

compared this modified shape of Mitre Peak against the original SRTM and other freely available DEMs (see section 3.5).

### 3.2. Workflow for estimating trends in glacier elevation change in shaded areas

We created a binary mask of shaded and non-shaded areas (Fig. 3a) by applying a user-defined threshold to the digital numbers of the green band (encompassing a wavelength of 525-600 nm) of each Landsat scene (Appendix A1). We found the

green band useful because shadows appear dark on the otherwise bright glacier surface. Snow, firn, and ice have minimal absorption in the blue-green range, whereas red and infrared light is strongly absorbed on these surfaces. This trait enhances contrast at the interface of glaciated surfaces and shaded, colder areas with increasing wavelength. Incoming and reflected electromagnetic wavelength in the green band is also less affected by the Rayleigh scattering in the atmosphere compared to the blue band that has a shorter wavelength. The green band therefore offers a good compromise between contrast and surface

reflectance measured at the sensor, and has been successfully used in mapping glacier outlines (Paul et al. 2016). For each Landsat image, we obtained the sun azimuth and sun elevation from the associated metadata file. We used these two parameters to simulate cast shadows using a ray-tracing algorithm implemented in SAGA-GIS V2.3.2 (Conrad et al. 2015). This algorithm returns a binary raster classifying each pixel either as shaded or non-shaded, equivalent to our threshold-based mapping (Fig. 3b). We then calculated the difference in area between the modelled shadow and shadows derived from Landsat images.

We clipped the resulting polygons to the glacier outline in the Randolph Glacier Inventory (RGI) V6.0 (Pfeffer et al. 2014) (Fig. 3c). Within these difference polygons, we obtained the change in shadow length using bearing lines at a regular horizontal spacing of 30 m (i.e. the cell size of Landsat images) in the direction of the sun azimuth (Fig. 3d–f, Appendix A1). These lines

represent the incoming sun rays and are assumed to be parallel, given that the Sun is a far distant, point-shaped light source. Thus, the change in shadow length is considered relatively short compared to the distance between Earth and Sun. Artefacts

in the bearing lines (Fig. 3e) appeared mainly because of the limited resolution of the DEM and satellite images (i.e. interrupted lines by pixel corners, shadows at the bottom edge or in ice-free areas of the glacier), so we removed them manually. Finally, we used the trigonometric relationship of the law of tangents to convert the length of each line to changes in elevation relative to the date when the DEM was acquired (Fig. 1). Earth curvature could influence the length of the simulated shadows and thus the glacier elevation changes, albeit only in the millimetre range, and is therefore not considered in our analysis.

We scaled the elevation changes for each glacier so that the median for the year 2000 is zero, because in most cases the data are relative to the elevation values in the SRTM DEM from February 2000. The changes in glacier elevation in the other years are therefore the positive or negative deviations from the median in 2000.

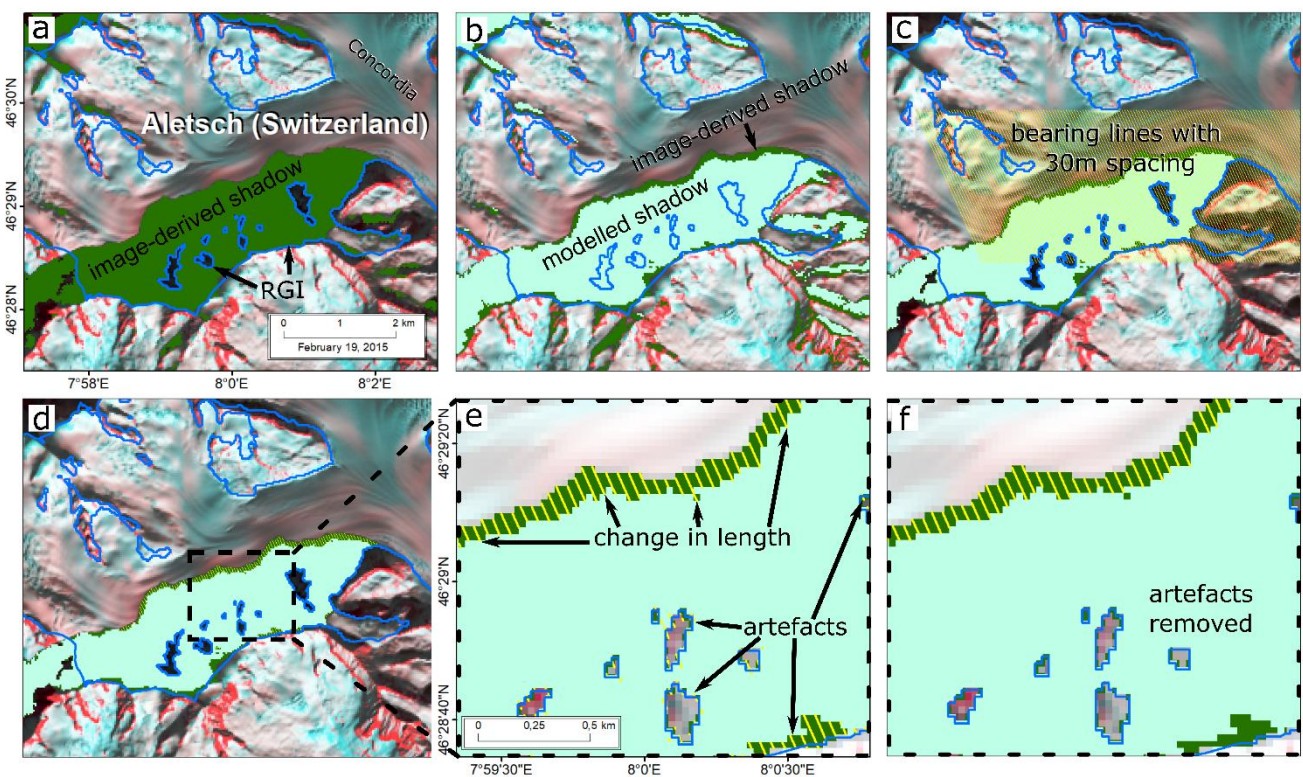

**Figure 3: Flowchart of modelling terrain shadows using the example of Great Aletsch glacier. (a)** Mapped shadows (green) using a
threshold of 5,500 in a Landsat 8 image. Background image is a false-color composite using the SWIR, blue, and green bands, and the glacier outline according to the RGI V6.0 is blue. **(b)** Modelled shadow (turquoise) using SAGA-GIS, draped over the mapped shadow in the Landsat image. **(c)** Extracted shadows by RGI and parallel bearing lines from the azimuth given in the Landsat metadata. **(d)** Lines cut to the difference between the two shadows. **(e)** Close up of **d** with generated lines of change in shadow length and unwanted artefacts. **(f)** Artefacts at the bottom edge and along ice-free areas are removed.

165        We used a Bayesian multi-level linear regression model to estimate linear trends in elevation change for each glacier with time. Multi-level models can accommodate groups in data, in our case different glaciers, within a single model. We can thus estimate local effects at a given glacier with respect to the entire population learned from all data regardless of their location. Multi-level models improve parameter estimates for individual groups, in particular when differing sample sizes cause variance across the groups (McElreath 2020). Multi-level models are suitable for datasets with different sample sizes in

each group. In our case, one glacier might have hundreds of bearing lines in a given year (e.g. Great Aletsch Glacier) and others might have fewer data (e.g. Gulkana Glacier, regarding the eastern shadow). The hierarchical model structure avoids over-fitting parameters for glaciers with many bearing lines and generally improves inference for groups with few data points. The glaciers inform each other, given that groups are conditioned on the data from all glaciers, reducing uncertainty in years with few bearing lines at a given glacier. The parameters in the model are drawn from distributions specified by population-

level (hyper-)parameters, which are also learned from the data. The multi-level model returns the posterior distribution for both population-level and group-level parameters.

       Our likelihood function follows a Student's $t$-distribution, which is robust against outliers (Kruschke 2014). We modelled the trend in glacier elevation change $\Delta h$ with year $y$ as

$$\Delta h_{ji} \sim t(\mu_{ji}, \kappa, \nu), \text{ for } j = 1, \ldots, J \text{ and } i = 1, \ldots, n_j \tag{1}$$

$$\mu_{ji} = \alpha_j + \beta_j y_{ji}, \text{ for } j = 1, \ldots, J \text{ and } i = 1, \ldots, n_j \tag{2}$$

$$\begin{bmatrix} \alpha_j \\ \beta_j \end{bmatrix} \sim MVNormal \left[ \begin{pmatrix} \alpha \\ \beta \end{pmatrix}, S \right] \tag{3}$$

$$S = \begin{pmatrix} \sigma_\alpha & 0 \\ 0 & \sigma_\beta \end{pmatrix} R \begin{pmatrix} \sigma_\alpha & 0 \\ 0 & \sigma_\beta \end{pmatrix} \tag{4}$$

$$R = \begin{pmatrix} 1 & \varsigma \\ \varsigma & 1 \end{pmatrix} \tag{5}$$

where $\Delta h$ are the elevation changes from bearing lines in each year, $i$ is an index for $n$ bearing lines, and $J$ is the number of

glaciers. The likelihood function has a location parameter $\mu$, $\kappa$ is a positive scale parameter, and $\nu$ are the degrees of freedom, fixed at $\nu = 3$. The parameters $\alpha_j$ and $\beta_j$ are the intercepts and slopes for each group, respectively, and $\alpha$ and $\beta$ are the corresponding parameters on population-level. The covariance matrix $S$ is composed of group-level standard deviations $\sigma_\alpha$ and $\sigma_\beta$, and $R$, the correlation matrix with correlation $\varsigma$. We choose the following priors to model the parameters for the entire population and all groups (i.e. the glaciers)

$$\kappa \sim N(0, 2.5) \tag{6}$$

$$\alpha \sim N(0, 2.5) \tag{7}$$

$$\beta \sim N(0, 2.5) \tag{8}$$

$$\sigma_\alpha \sim N(0, 2.5) \tag{9}$$

$$\sigma_\beta \sim N(0, 2.5) \tag{10}$$

$$R \sim LKJCholesky(1). \tag{11}$$

These priors refer to standardised data pairs ($\Delta h$ and $y$) with zero mean and unit standard deviation. Choosing wide priors with a zero-mean Gaussian and standard deviation of 2.5 admits both negative and positive trends for $\beta$, such that the posteriors are largely informed by the data. We choose a Lewandowski–Kurowicka–Joe (LKJ) Cholesky correlation distribution prior for $R$, so that all correlation matrices are equally likely. We numerically approximate this posterior using a
Hamiltonian sampling algorithm implemented in Stan that is called via the software package brms within the statistical programming language R (Bürkner 2017; R Core Team 2022; Stan Development Team 2022). We ran three parallel chains with 6,000 iterations after 2,000 warm-up runs, and found that the Markov chains have converged ($\hat{R}$ statistic = 1.0). We report the posterior distributions of all model parameters in Table A3.

**3.3 Comparison to reference DEMs and historical maps**

We compared our estimated trends in glacier elevation change with trends from independent multi-temporal, high-resolution DEMs in shaded areas. For all glaciers in North America, we used repeated DEMs available for USGS benchmark glaciers (McNeil et al. 2022). These DEMs have spatial resolutions ranging between a few decimeters to 10 m, and were derived from historic topographic maps, aerial stereo-photography, and space-borne imagery. For all DEMs, we extracted the mean elevation change of all pixels between the edges of the largest and smallest mapped shadows in the Landsat images, as
the shape of the shadows varies due to changing acquisition dates and sun angles. For the Great Aletsch Glacier, we obtained glacier elevation changes from historical maps (Landeskarte at map scales of 1:25,000 and 1:50,000) available for 12 years between 1959 and 2020 from the Bundesamt für Landestopografie KOGIS (Koordination, Geoinformation und Services, https://www.swisstopo.admin.ch, last access: 26. March 2023). Mountain peaks in these maps are labelled with elevation values and we consider them as stable terrain in the past 60 years. A sample of 10 peaks suggests positive and negative offsets
of less than 5 m compared to the high-resolution SwissALTI3D DEM, making these maps suitable for validating our method over a period of more than six decades (Fig. A1; Table A5). To infer elevation changes from contour lines in historical maps, we manually chose four points with a spacing of 1 km along a straight line in the flow direction of the glacier within the area covered by the shaded glacier (Fig. A1). For each map, we then extracted the glacier elevation at each point using linear interpolation and calculated the average elevation change from these points. We could not find any historical elevation data
for the Baltoro Glacier that would be suitable for comparison.

We used the same multi-level structure as above (Eqs. 1-11) to determine the trends in glacier elevation change from glaciers with repeat, high-resolution DEMs. To this end, we conditioned the model on $J = 5$ glacier shadows (excluding Baltoro), chose the same priors, and maintained the setup of the Hamiltonian sampler. We learned two models, one with all available data and one with data limited to the Landsat period, to make trends comparable to our study period. In either case, we found that all chains converged ($\hat{R} = 1.0$) and report all model parameters in Table A4.

## 3.4 Comparison to glacier elevation changes from Hugonnet et al. (2021)

In addition, we compared the elevation changes of our six study glaciers with time series from Hugonnet et al. (2021). In their study, the entire archive of satellite images from the Advanced Spaceborne Thermal Emission and Reflection Radiometer (ASTER) mission was automatically assembled into DEMs, stacked and co-registered with other DEMs from the ArcticDEM at a spatial resolution of 100 m x 100 m. In general, each pixel is covered by several dozen DEMs over the period 2000 and 2019. Noise and artefacts in the DEMs that would lead to excessively strong rates of glacier elevation change are iteratively filtered from the time series by several fixed thresholds, deviations from the reference TanDEM-X DEM, as well as by a Gaussian Process (GP) regression model. Unlike our linear regression model, the GP regression model allows for seasonal, periodic oscillations in glacier elevation, so that the interpolated time series of glacier elevations show seasonal variations. We used time series of glacier elevation change extracted from the area between the largest and the smallest shadow (Fig. 2), provided as summary statistics on mean glacier elevation change between 2000 and 2019 by Romain Hugonnet (pers. comm., 2023) (Fig. 6). For comparison, we shortened our study period to 2000-2019 and fitted the Bayesian hierarchical model with the same structure and parameterisation as above.

## 3.5 Sensitivity of cast shadows against globally available DEMs

The choice of the DEM may bias our estimates of glacier elevation changes because the DEMs can have different spatial resolutions, artefacts, and horizontal and vertical errors, e.g. due to foreshortening, layover, and shadow effects in radar data. These uncertainties propagate into modelled cast shadows and likely change the inference on glacier elevation change derived from different globally available DEMs (Table A2). Using the Great Aletsch Glacier and the Baltoro Glacier, we quantitatively and qualitatively assessed the impact of the underlying DEM on the modelled shadows.

The Great Aletsch Glacier provides seven freely available DEMs. From Open Topography (https://opentopography.org/), we obtained two SRTM DEMs (SRTM-1 with 30 m and SRTM-3 with 90 m spatial resolution), the NASADEM (a reanalysis of SRTM data with 30 m resolution), ALOS World 3D (AW3D30 with 30 m), and two Copernicus DEMs (GLO-30 with 30 m and GLO-90 with 90 m). We compared the DEM-derived shadows to those from the LiDAR-based swissALTI3D DEM, which we treat as the benchmark. In each simulation, we use a sun azimuth of 135° and sun elevation of 25° to determine how much the modelled shadow varies in shape and extent as a function of the input DEM.

Accordingly, more recent DEMs should generate longer shadows, if the glacier has gradually thinned during that period. We also studied the role of the DEM resolution on Landsat-derived shadows at Great Aletsch Glacier. In theory, choosing a DEM

resolution coarser than Landsat (30 m) could increase the noise in the bearing lines, as one DEM pixel would cover several Landsat pixels, and thus several bearing lines. To test this idea, we calculated the difference between the shadow mapped from Landsat images and the shadow simulated from three input DEMs. We then compared the variance of elevation change with time using bearing lines drawn through the swissALTI3D DEM (5 m, highest resolution), the SRTM-1 (30 m, medium resolution, corresponding to that of the Landsat images), and the GLO-90 DEM (90 m, lowest resolution).

The example of the Baltoro Glacier addresses the impact of the unknown elevation of Mitre Peak on the size and shape of the cast shadow. The SRTM data has gaps so that the peak is not well represented by the data, and the 8-m HMA DEM has wide data gaps on the west-facing slopes of Mitre Peak (Shean 2017). Therefore, we took the void-filled SRTM-1 and SRTM-3 products as a basis, cut out Mitre Peak from these DEMs, and inserted the raster values from AW3D30, NASADEM, COP30, COP90 and VFP for the peak. We then mapped the shadow from a Landsat image obtained in February 2000 (the

acquisition period of SRTM), and compared its shape against a modelled using these modified input DEMs and the azimuth and elevation angle from the Landsat image. We assume that the DEM with the smallest differences between modelled and mapped shadows is most suitable to represent mountain peaks, and thus elevation changes.

## 4 Results

### 4.1 Glacier elevation changes from cast shadows

In each Landsat scene, $31^{+79}/_{-11}$ bearing lines (median, 2.5%, and 97.5% of the distribution) with a regular spacing of 30 m between each line pass through the mapped shadows on the five selected glaciers (Appendix A1). Individual bearing lines suggest the lowest variance in glacier elevation change at Sperry Glacier (-22 m to +5 m; 2.5% and 97.5% of the distribution) and the highest variance at Gulkana West (-94 m to +30 m), when adjusting elevation changes relative to the year 2000. Our analysis of trends in glacier elevation changes suggests that Gulkana West and Great Aletsch Glacier had the highest

annual rates of thinning of $-1.21^{+0.15}/_{-0.16}$ and $-1.08^{+0.05}/_{-0.05}$ m yr$^{-1}$, respectively (mean and 95% highest density interval, HDI). The mean elevation change in the western, lower-lying arm of Gulkana Glacier is about 10 times more negative than that of the eastern, higher-lying arm. Sperry and South Cascade Glacier lost on average about 0.4 m per year since the late 1980s. The eastern arm of the Gulkana glacier has been thinning at a credible negative, albeit low, annual rate, while the surface of the Baltoro Glacier had no change in recent years (Fig. 4).

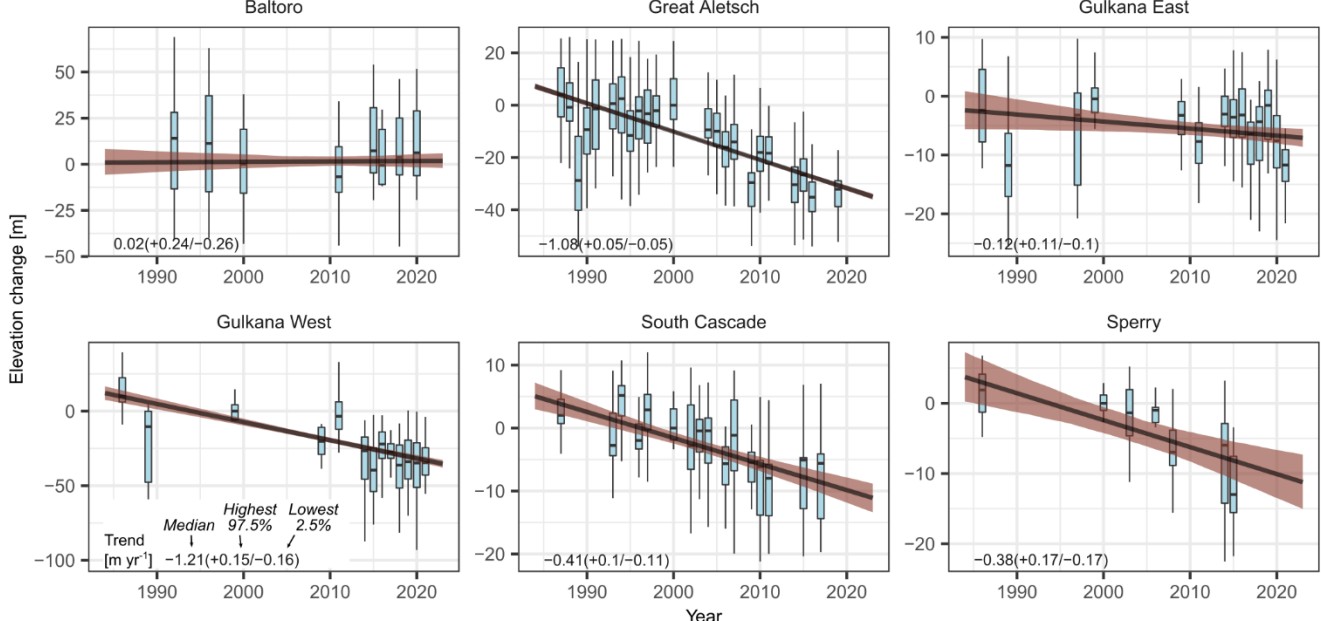

**Figure 4: Trends in mean elevation change on shaded glacier surfaces.** Boxplots show annual glacier elevation changes, which we have derived from bearing lines drawn through shadows in Landsat images. Values of elevation change are relative to the median value in 2000 (for Gulkana in 1999). Boxes encompass the interquartile range, whiskers are 1.5 times the interquartile range, and horizontal lines are the median. Outliers (lowest and highest percent in the distribution) are removed. Thick black line is the mean posterior trend and brown shade is the 95% highest density interval (HDI). Numbers in lower left corner summarise the posterior distribution of the trend in glacier elevation change, including the median, the lower 2.5%, and the upper 97.5% of the HDI.

## 4.2 Comparison with reference DEMs

Our estimated trends from bearing lines match the trends obtained from high-resolution DEMs and historical maps (Fig. 5). However, uncertainties in the trends calculated from the reference DEMs are consistently higher given that fewer data enter the hierarchical regression model, especially if we fit the model only to data obtained during the shorter Landsat period. At Great Aletsch Glacier, we find similar trends in mean glacier elevation change between our method ($-1.08^{+0.05}/_{-0.05}$ m yr$^{-1}$) and the reference DEMs both since 1959 ($-1.06^{+0.27}/_{-0.31}$ m yr$^{-1}$) and during the Landsat period ($-0.88^{+0.49}/_{-0.76}$). At South Cascade Glacier, the mean trend from the high-resolution DEMs is more than twice that of the trends obtained from bearing lines ($-1.06^{+0.54}/_{-0.45}$ vs. $-0.41^{+0.1}/_{-0.11}$ m yr$^{-1}$). Trends are more consistent, however, if we consider all available data from South Cascade, extending back to late-1950s (Fig. 5). For the two shadows at the Gulkana glacier, the mean trends from the DEMs during the Landsat period are negative and midway between the very high and low values that we had determined for the two

arms. Trends in historical DEMs are difficult to determine at Sperry Glacier because only two observations inform the multi-level model during the Landsat period.

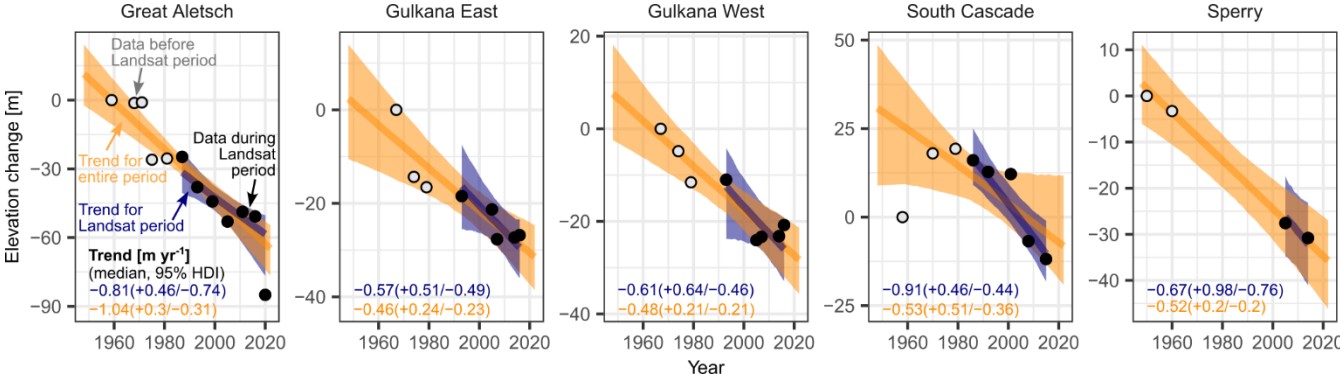

**Figure 5: Reported glacier elevation changes in five shadowed areas on four glaciers.** All values are relative to the first observation for each glacier, which is set to zero. Black bubbles are observations when Landsat images are available for a given glacier (see trends in Fig. 4). Grey bubbles mark data obtained before the Landsat period. Shades, thick lines, and numbers refer to models fit to all data from the entire period (orange), and to data for the Landsat period only (blue). Numbers in lower corner left summarise the posterior distribution of the annual trend in glacier elevation change, including the median, the lower 2.5%, and the upper 97.5% of the HDI.

### 4.3. Comparison with data from Hugonnet et al. (2021)

If we reduce our study period to 2000-2019, we find that our trends generally follow those of Hugonnet et al. (2021) (Fig. 6). The exception is Gulkana East, where our estimated mean rate of glacier elevation change is twice as high. The most negative trends in both methods occurred at the Great Aletsch Glacier. With the exception of one year on the Great Aletsch and Gulkana East glaciers, the Gaussian process regression models of Hugonnet et al. (2021) overlap with our data (yellow interquartile ranges of the boxes in Fig. 6), indicating good agreement between the two methods. One reason for some of the discrepancy between the two datasets may be the rigorous filtering of outliers in the dataset of Hugonnet et al. (2021), whereas our method maintains the elevation changes of all bearing lines, regardless of their distances from the mean or median.

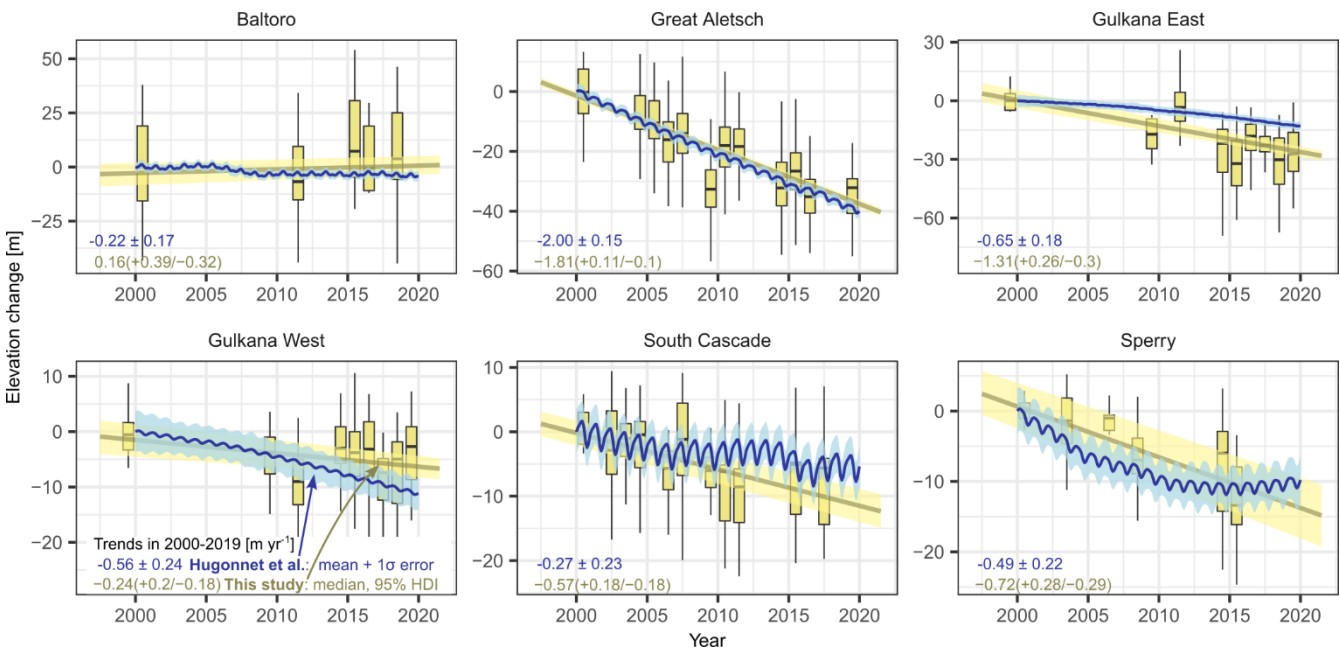

**Figure 6: Glacier elevation changes in shaded areas using our method and that of Hugonnet et al. (2021) for data between 2000 and 2019.** All values are relative to the year 2000, which is set to zero. Yellow colors refer to our method, and blue colors are trends of glacier elevation change using Gaussian Process (GP) regression through time series of ASTER DEMs from Hugonnet et al. (2021). Boxes encompass the interquartile range, whiskers are 1.5 times the interquartile range, and horizontal lines are the median. Outliers (lowest and highest percent in the distribution) are removed. Thick yellow line is the median posterior trend and light-yellow shade is the 95% highest density interval (HDI). Yellow numbers in lower left corner are our posterior estimate of the annual trend in glacier elevation change, including the mean, the lower 2.5%, and the upper 97.5% of the HDI. Blue numbers are the mean annual trend and 1σ error from Hugonnet et al. (2021).

## 4.4 Influence of DEM type and resolution

We conducted the shadow-based detection of glacier elevation changes with three DEMs for the Great Aletsch (Fig. 7). The length of bearing lines between shadows (and derived elevation changes) scatters substantially, but the shapes of nonparametric regression curves are consistent between the different DEMs. Apart from these trends, residuals from these trends are affected by the underlying DEM. Residuals of the SRTM-1 and GLO-90 had a high standard deviation of 18.2 m and 26.8 m. Residuals are lowest for the swissALTI3D DEM at a standard deviation of 14.3 m, suggesting that an increase in DEM resolution may improve the precision of our method.

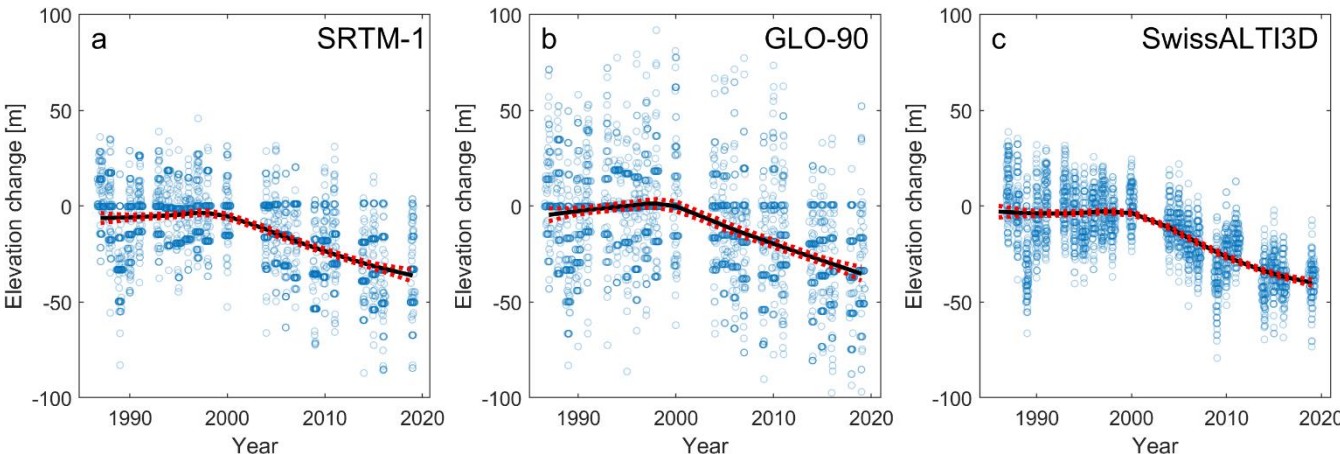

**Figure 7: Glacier elevation changes of the Aletsch Glacier (see extent in Fig. 3) based on Landsat imagery and modelled shadows derived from three digital elevation models (DEMs).** Semi-transparent blue points show the elevation change derived from the length of individual bearing lines between Landsat-derived shadows and those modelled from **a)** the 30-m SRTM-1 DEM, **b)** the 90-m GLO-90 DEM, and **c)** the 5-m swissALTI3D DEM. Black lines are the means from a lowess regression of elevation change against time. Dashed red lines are bootstrapped confidence intervals ($\pm 2\sigma$).

## 4.5 Comparison of shadows derived from DEMs

The elevation changes obtained from bearing lines have substantial variance in a given year (Fig. 4), despite covering a small range in elevation along the glacier. We infer that DEM resolution and quality have important controls on estimated glacier elevation changes from cast shadows. Indeed, the example of the Great Aletsch Glacier shows that different DEMs produce shadows of different lengths, even with constant sun azimuth and elevation (Figs. 8, 9). This variation reflects limits in the DEM resolution and the representation of ridge lines. The acquisition date may also play a role, assuming that ongoing thinning might produce longer shadows in more recent DEMs. In our example, shadows projected from swissALTI3D DEM (5 m spatial resolution, acquisition in 2017 and 2018) extend farthest to the north (Fig. 8a). The large shadow area thus likely follows both from the reported decadal glacier thinning and from a more precise representation of the ridge line and the surrounding topography (Fig. 8a). Shadows from the GLO-30 DEM (acquisition date 2011-2015, ~30 m spatial resolution) are very similar to those derived from the swissALTI3D DEM (Figs. 8b, 9). We also find the smallest variance in shadow length for the GLO-30 DEM (Fig. 9). Shadows derived from the GLO-90 DEM (~90 m resolution) show both a larger spatial offset (Fig. 8c) as well as a higher variability in shadow length (Fig. 9). We attribute this mismatch to a higher degree of spatial averaging, causing lower topographic ridges due to the coarser spatial resolution. Shadows derived from the AW3D30 DEM (acquisition period between 2006 and 2011, ~30 m spatial resolution) are highly variable compared to the swissALTI3D DEM (Fig. 8d). Some of the shadows extend beyond those derived from the swissALTI3D DEM, an effect of exaggerated topography in the DEM that overestimates the height of the ridge (Fig. 9). Finally, shadows derived from the SRTM DEMs and

365 NASADEM (Fig. 8e-g) – all derived from data acquired from the same shuttle mission in 2000 – show the highest difference to the swissALTI3D DEM. SRTM DEMs and NASADEM derived shadows are very similar, but again, the coarser SRTM-3 DEM leads to a lowering of the ridges and larger horizontal distances.

Absence of high-resolution data and voids in the SRTM data covering Baltoro Glacier and Mitre Peak prompted us to use the VFP DEM to obtain the shape of the steep and peaked mountain. We assume that the unknown acquisition date of 370 topographic data in the VFP DEM has little impact on our subsequent analysis as Mitre Peak is free of glacier ice and no major rockfalls were reported during our study period that could have reduced its elevation. To evaluate this choice, we compared the modelled shadows based on elevation data of the Mitre Peak obtained from all DEMs with the actual shadows cast by the mountain in 2020 (Fig. 10). The VFP DEM has a spatial resolution of 3 arcseconds (90 m) which suggests that it will perform less well than the other DEMs with higher resolution. However, visual comparison shows that the VFP DEM captures the 375 actual shadows more precisely, which is consistent with >100 m higher peak elevations than those contained in the other DEMs (Fig. 10).

In summary, variations in modelled shadows obtained from different DEMs relate to variable acquisition dates but also reflect how accurately ridge topography is represented in the DEMs. Comparison of DEMs with the same acquisition date but different spatial resolution show that coarser DEMs underestimate ridge height and commensurately shadow length. 380 Notwithstanding, a general trend towards longer shadows and thus a trend towards lower glacier elevations can be observed for younger acquisition dates (Fig. 9).

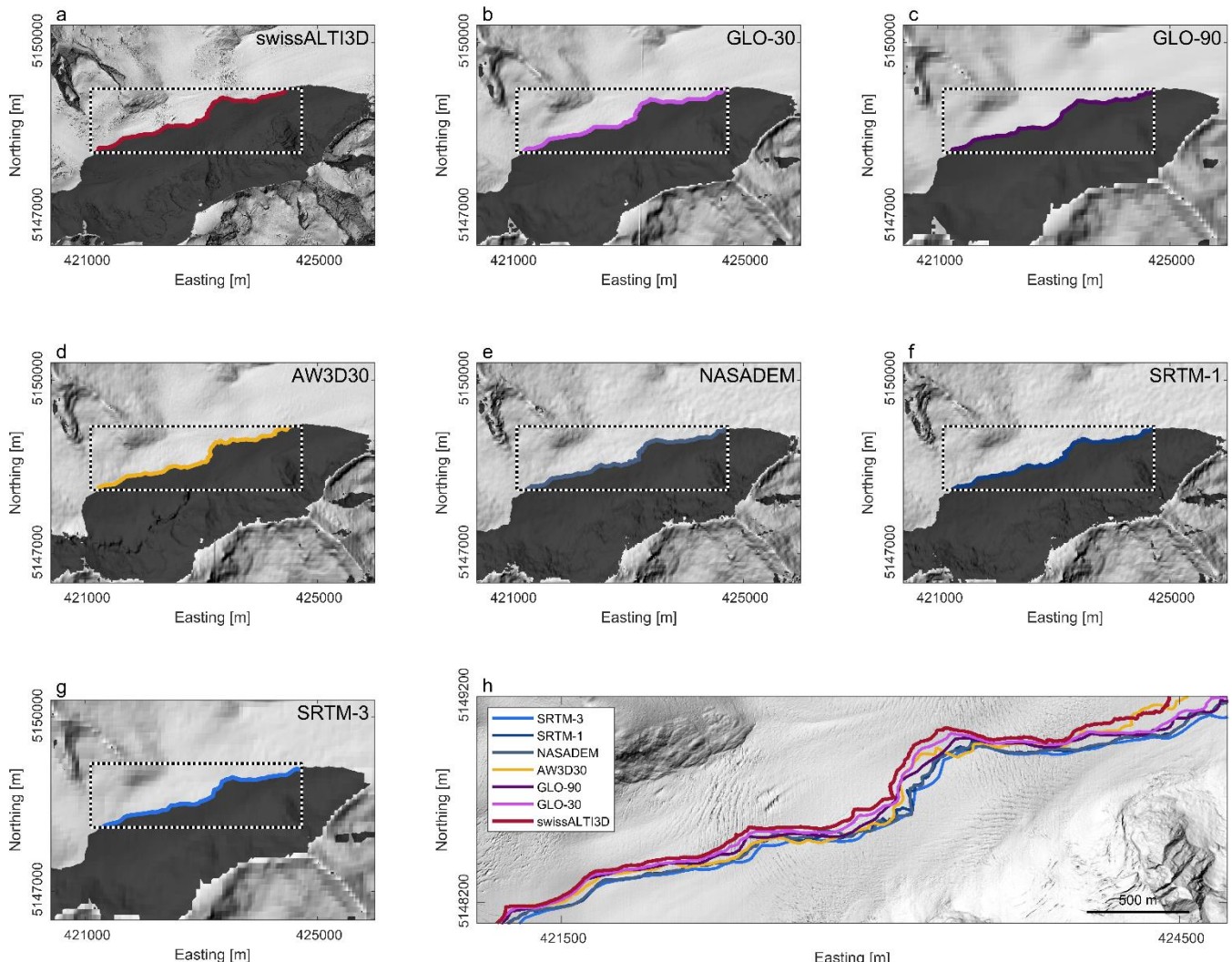

**Figure 8: Shadows projected onto Great Aletsch Glacier using different digital elevation models. (a-g)** Grey hillshades show the simulated cast shadow using a sun azimuth of 135° and elevation of 25°. **(h)** Close-up of the shadow outlines modelled with different DEMs. Hillshade in the background is from the swissALTI3D DEM.

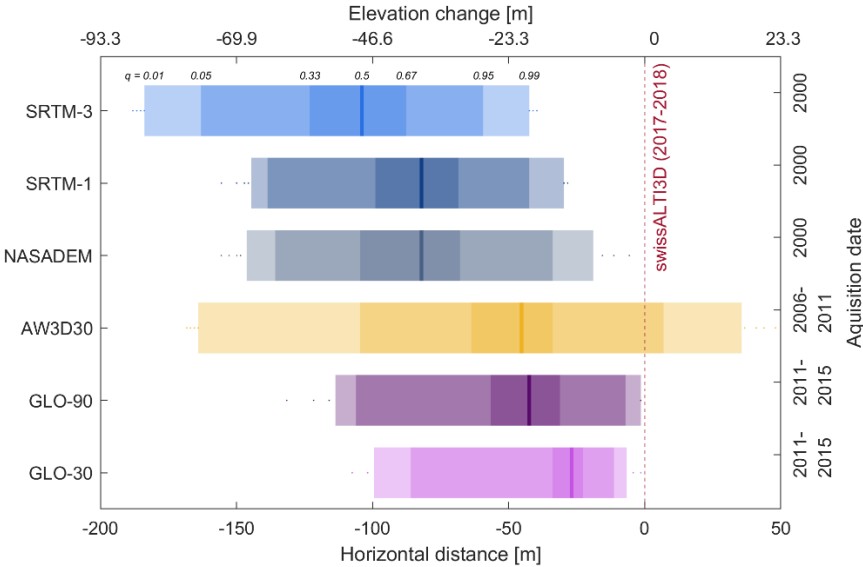

**Figure 9: Difference in the lengths (and corresponding elevation changes) of bearing lines crossing a shadow on Great Aletsch using six DEMs and the benchmark swissALTI3D DEM.**

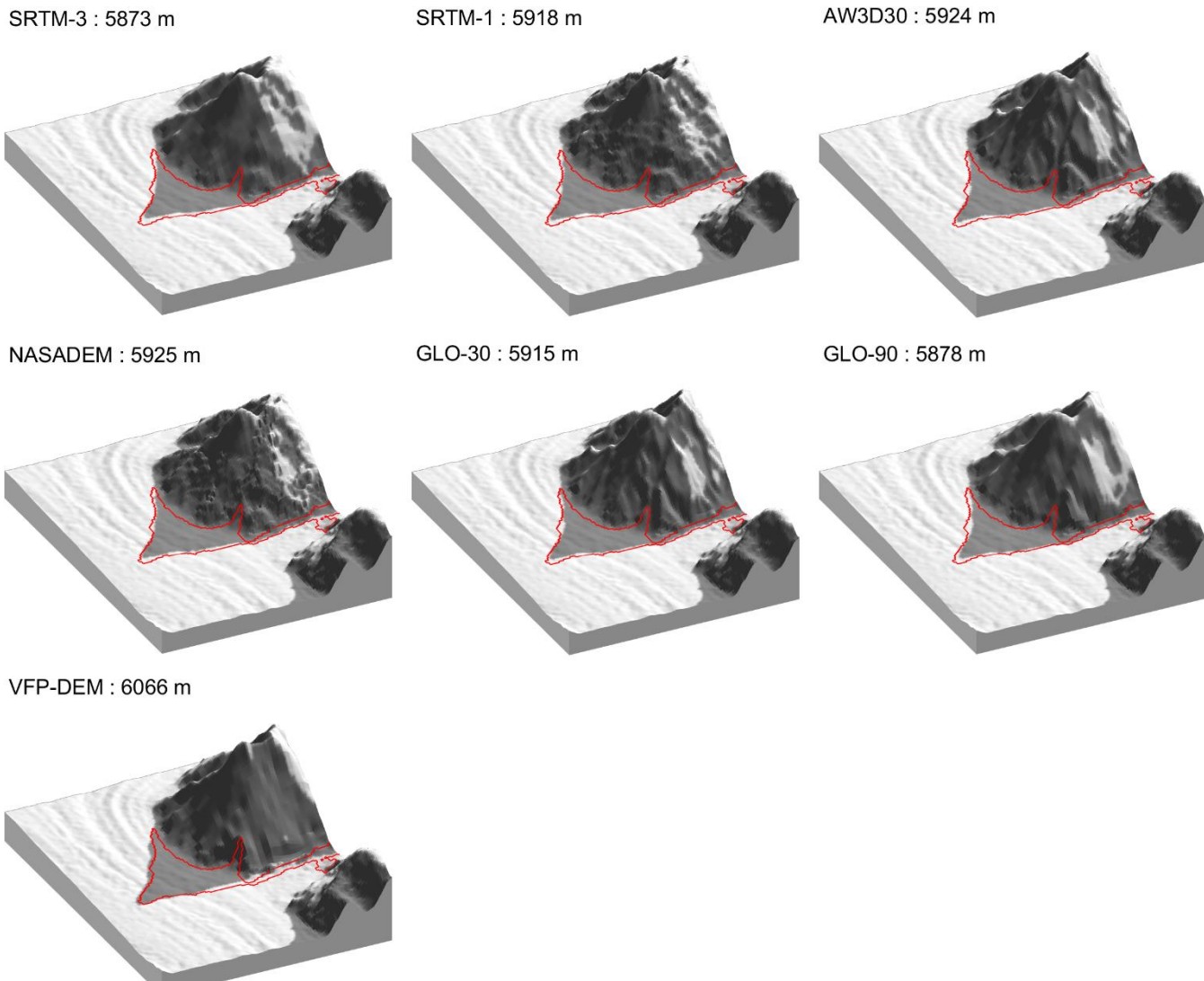

390

**Figure 10: 3D surface views including modelled and mapped shadows cast from Mitre Peak onto Baltoro Glacier, Pakistan.** We used the SRTMGL1 and replaced the Mitre Peak with elevation data from different DEMs. Shadows were calculated with an azimuth of 151.9° and a sun elevation angle of 29.5°. These values refer to the sun position during the acquisition time (Jan 24, 2000) of the Landsat image from which shadows were mapped (red outline). Visual comparison shows that the VFP-filled SRTM-1 creates the best match between 395 modelled and actual cast shadows of the peak, whereas there are pronounced offsets between actual shadows and those derived from other DEMs. Elevation values in labels indicate the peak altitude of Mitre Peak in each DEM.

## 4 Discussion

We developed and assessed a method that measures glacier elevation changes in remote areas based on cast shadows from adjacent mountains. The precision and accuracy of the method depend on several factors that pertain to the individual processing steps and the input data (Rada Giacaman 2022). We show that DEM quality and resolution cause variability in the detected elevation changes (Figs. 7-9). To this end, we assessed the length of bearing lines that link the shadow outlines along the azimuth direction. We find that spatial resolution affects the precision and accuracy of these lines (Fig. 7). First, DEMs with coarser resolution decrease the precision due to spatial averaging, blurring ridge topography by smoothing peaks and saddles (Purinton and Bookhagen 2017). This effect may be more pronounced in SRTM data, which can have high errors on steep slopes and often poorly represent ridges and valley bottoms (Gorokhovich and Voustianiouk 2006; Schwanghart and Scherler 2017). Coarser resolution also biases, or decreases the accuracy of our estimates because DEM values along ridges are lowered by spatial averaging (Fujita et al. 2008). Both effects entail that modelled shadow outlines on glaciers increasingly lack detail and underestimate shadow length with coarser DEM resolution (Fig. 9). Poor quality of the underlying DEM will propagate into estimates of glacier elevation change although trends derived from different DEMs are surprisingly consistent (Fig. 7). Satellite imagery obtained for the date of DEM acquisition can help quantify and correct for such biases.

Besides differences in resolution, the type of DEM also impacts the precision and accuracy of modelled shadows. Our analysis shows that among the DEMs with global coverage, the new GLO-30 DEM has the highest precision of derived shadows when compared to the benchmark swissALTI3D DEM (Fig. 9), which is consistent with recent DEM assessments that underscore the high performance of the GLO-30 DEM (Guth and Geoffroy 2021). Shadow outlines calculated from NASADEM and SRTM-1 are similar as they are obtained from the same data. We acknowledge that our method leaves any effects of SAR penetration into the snow pack covering the glacier ice (Berthier et al. 2006) unconsidered. Yet, this offset can be treated as a constant when drawing bearing lines through shadows, given that the input DEM (SRTM) remains unchanged in our analysis. Snow cover can be thick in accumulation areas and may lead to biases (underestimates) when calculating glacier volume changes from DEM differencing (Gardelle et al. 2012). Though most shadows in our cases are in ablation zones, we recommend to account for differing penetration depth in future studies that also include shadows on glaciers at very high elevations in snow accumulation areas. The relatively low performance of the AW3D30 DEM in comparison to other global DEMs likely relates to hillslope and ridge artifacts caused by errors in optical DEM generation (Purinton and Bookhagen 2017). Where steep topography severely impacts DEM quality, manually edited DEMs such as the VFP DEM can provide a viable alternative despite their relative coarse spatial resolution. In any case, our Bayesian framework objectively propagates these errors and uncertainties. One promising avenue for future research is to use more informed priors based on previous research on glacier elevation change (Hugonnet et al. 2021). Narrower and stronger priors may reduce the width of our posterior trends on glacier elevation changes that we currently observe at Sperry Glacier, for example (Fig. 4). They might also offer a better compromise to balance some of the differences within our data (e.g. between Gulkana East and West), and also between

our data and data from previous research. One of these examples may be the outstanding trend at Gulkana West (Fig. 6), where the physical causes and methodological differences between our appraisals and that of Hugonnet et al. (2021) remain to be determined.

In addition to the resolution and quality of the DEM, we expect that higher image resolution will warrant a higher accuracy and precision at which elevation changes can be detected (Fig. 7). We refrained from analyzing the effects of image
resolution because we used only Landsat imagery, which are the longest freely available time series of satellite imagery. We recall that our trigonometric approach hinges on sun elevation and image resolution provided in the image meta data, both setting the detection limit of elevation changes. For example, for a sun elevation of 20° and a spatial resolution of 30 m, a minimum elevation change of 10.9 m can be detected, unless subpixel classification approaches or pan-sharpening techniques are adopted (Liu and Wu 2005). Sun angle will be critical for our method (Rada Giacaman 2022) and we expect that our
approach works better for images acquired during the winter months of the respective hemispheres as well as at higher latitudes. To determine interannual trends, we recommend using satellite imagery with similar time stamps within a year, given that glacier elevations are prone to seasonal variations (Moholdt et al. 2010).

Atmospheric refraction – the bending of solar light as it traverses the atmosphere – causes an apparently higher sun elevation. The offset between the actual and apparent solar-position leads to errors in shadow-height applications depending
mainly on solar elevation and, to a minor degree, on atmospheric pressure, humidity and temperature (Rada Giacaman 2022). Sun elevations in our study range between 15 and 40° which yields height difference errors of 0-2% (see Fig. 10 in Rada Giacaman, 2022). Additional error sources include uncertainties in the position of the satellite as well as problems in image registration and deformation. Yet, we did not account for errors due to atmospheric refraction and image registration as they appear minor compared to those related to image resolution and DEM quality.

Our study reveals and confirms decadal-scale loss of glacier mass. These changes are consistent with independent estimates of glacier elevation changes based on stereo-photogrammetric analysis of US benchmark glaciers, i.e., South Cascade, Gulkana and Sperry Glacier (McNeil et al. 2022), and historic topographic maps of Great Aletsch Glacier (Fischer et al. 2015; Leinss and Bernhard 2021). For the Baltoro Glacier, we detect no credible trends and independent, field-based validation data of surface changes at the shadow location are lacking. Yet, comparison of photographs from 1909 and 2004
show that glacier elevation changes at Concordia were low in the 20[th] century (<40 m) (Mayer et al. 2006). These small rates of surface lowering have been attributed to increases in precipitation and a lowering of summer mean and minimum temperatures in the Karakoram, supporting regionally unchanged glacier masses referred to as 'Karakoram Anomaly' (Hewitt 2005; Kääb et al. 2015; Forsythe et al. 2017; Farinotti et al. 2020).

We stress that our results are tied to local changes of shadows casted from adjacent mountains. Thus, we caution
against comparing our results directly with glacier-wide mass balances because these integrate over entire glaciers or elevation bands within glaciers, and may refer to different study periods. For example, Hugonnet et al. (2021) estimates that the entire

areas of Great Aletsch and South Cascade Glacier had elevation changes of -1.42 ± 0.1 and -0.66 ± 0.15 m yr$^{-1}$ (mean and 1$\sigma$ error), respectively, in 2000-2019. Our estimates are less negative (-1.08$^{+0.05}$/$_{-0.05}$ and -0.57$^{+0.17}$/$_{-0.17}$ m yr$^{-1}$, respectively) in the longer Landsat period, either because we measure elevation changes at higher parts of the glacier with possibly lower melt

rates, or because glacier melt has accelerated in recent years (Hugonnet et al. 2021). Indeed, if we shorten the study period to the years 2000-2019, the Great Aletsch Glacier shows a trend of elevation change that is almost twice as high as for the much longer trend going back to the 1980s. We thus envision that our method could enhance, complement, and amend geodetic surveys in ablation and accumulation areas (Beedle et al. 2014). Our method can be applied globally, but is restricted to those glaciers that are surrounded by stable topography. Our method becomes unsuccessful when the shadow edge constantly falls

onto bedrock due to progressive glacier retreat – a situation that will soon occur at the dwindling Sperry Glacier for example. Ideal environments for our approach are glaciers close to steep topography in high latitudes, producing cast shadows long enough to infer differences in bearing lines. Suitable sites remain to be identified and should, at best, have high-resolution DEMs with high precision and accuracy available. Locations with large landslides that lower mountain peaks (Shugar et al. 2021) should be avoided as they may violate the assumption of unaltered ridge topography over time. The processing steps

developed in this study can be fully automated although quality control of the obtained bearing lines connecting modelled and actual shadow outlines are crucial.

**5 Conclusions and outlook**

        In summary, we show that cast shadows offer avenues to retrieve glacier elevation changes from satellite imagery over many decades. We demonstrate for select cases that our method provides quantitative information about local changes in

glacier elevation with time. These changes are consistent with independent DEMs of difference in shaded areas. Accurately resolving glacier elevation changes hinges on the spatial resolution of the satellite imagery from which we mapped shadows, as well as the quality and resolution of the underlying DEM. Upon the emergence of global, void-free, high-resolution satellite images and DEMs, our method can be extended to historical satellite and aerial imagery, assuming that the geometry of mountain ridges and peaks remains unchanged with time. We conclude that our approach has the potential to complement

existing or future in-situ measuring networks anywhere on Earth where mountains shade parts of adjacent glaciers. We thus enrich the glaciological and geodetic toolbox with a new method that helps quantifying glacier elevation changes especially at high altitudes with limited access.

# Appendix A: Additional tables

**Table A1: Landsat bands used to map shadows on glaciers, including image metadata, the threshold to manually classify shadows on glaciers, and the number of bearing lines that cross shadows on glaciers. TM: Thematic Mapper, ETM+: Enhanced Thematic Mapper, OLI: Operational Land Imager, SRTM-1: DEM of the Shuttle Radar Topography Mission (30m resolution), swissALTI3D: DEM of Switzerland (5 m resolution), GLO-90: Copernicus DEM (90m resolution)**

| Glacier | Acquisition Date | Landsat Mission | Band | Azimuth | Elevation | File Name in GeoTIFF format | Threshold between shadow and no-shadow | Number of bearing lines drawn |
|---|---|---|---|---|---|---|---|---|
| Great Aletsch | 06.02.1987 | TM | 2 | 146.96 | 21.95 | LT05_L1TP_195028_19870206_20170213_01_T1_sr_band2 | 5,500 | 106 (SRTM-1, swissALTI3D), 107 (GLO-90) |
| Great Aletsch | 18.02.1988 | TM | 2 | 147.49 | 26.38 | LT05_L1TP_194028_19880218_20180215_01_T1_sr_band2 | 5,500 | 106 (SRTM-1), 105 (GLO-90, swissALTI3D) |
| Great Aletsch | 11.02.1989 | TM | 2 | 148.32 | 25.39 | LT05_L1TP_195028_19890211_20180215_01_T1_sr_band2 | 5,500 | 105 (SRTM-1), 106 (GLO-90) |
| Great Aletsch | 07.02.1990 | TM | 2 | 146.88 | 22.34 | LT05_L1TP_194028_19900207_20180219_01_T1_sr_band2 | 5,500 | 105 (SRTM-1, swissALTI3D), 106 (GLO-90) |
| Great Aletsch | 01.02.1991 | TM | 2 | 147.64 | 20.64 | LT05_L1TP_195028_19910201_20180215_01_T2_sr_band2 | 5,500 | 106 (SRTM-1, swissALTI3D), 105 (GLO-90) |
| Great Aletsch | 06.02.1993 | TM | 2 | 147.11 | 22.21 | LT05_L1TP_195028_19930206_20180215_01_T1_sr_band2 | 5,500 | 106 (SRTM-1), 105 (GLO-90, swissALTI3D) |
| Great Aletsch | 25.02.1994 | TM | 2 | 144.21 | 28.10 | LT05_L1TP_195028_19940225_20180215_01_T1_sr_band2 | 5,500 | 100 (SRTM-1), 101 (GLO-90, swissALTI3D) |
| Great Aletsch | 21.02.1995 | TM | 2 | 142.356 | 25.60 | LT05_L1TP_194028_19950221_20180215_01_T1_sr_band2 | 5,500 | 102 (SRTM-1, swissALTI3D), 101 (GLO-90, swissALTI3D) |
| Great Aletsch | 21.02.1996 | TM | 2 | 142.07 | 23.06 | LT05_L1TP_195028_19960215_20180215_01_T1_sr_band2 | 5,500 | 102 (SRTM-1, GLO-90, swissALTI3D) |
| Great Aletsch | 01.02.1997 | TM | 2 | 148.58 | 21.19 | LT05_L1TP_195028_19970201_20180215_01_T1_sr_band2 | 5,500 | 108 (SRTM-1), 107 (GLO-90, swissALTI3D) |
| Great Aletsch | 20.02.1998 | TM | 2 | 148.90 | 27.88 | LT05_L1TP_195028_19980220_20180215_01_T1_sr_band2 | 5,500 | 105 (SRTM-1, swissALTI3D), 106 (GLO-90) |
| Great Aletsch | 10.02.2000 | TM | 2 | 149.53 | 24.20 | LT05_L1TP_195028_20000210_20171211_01_T1_sr_band2 | 5,500 | 106 (SRTM-1, GLO-90, swissALTI3D) |
| Great Aletsch | 14.02.2004 | TM | 2 | 150.35 | 25.93 | LT05_L1TP_194028_20040214_20180311_01_T1_sr_band2 | 5,500 | 107 (SRTM-1), 106 (GLO-90), 108 (swissALTI3D) |
| Great Aletsch | 07.02.2005 | TM | 2 | 153.04 | 24.65 | LT05_L1TP_195028_20050207_20180130_01_T1_sr_band2 | 5,500 | 107 (SRTM-1), 108 (GLO-90), 109 (swissALTI3D) |
| Great Aletsch | 10.02.2006 | TM | 2 | 153.75 | 25.81 | LT05_L1TP_195028_20060210_20180311_01_T1_sr_band2 | 5,500 | 107 (SRTM-1), 109 (GLO-90, swissALTI3D) |
| Great Aletsch | 22.02.2007 | TM | 2 | 153.49 | 29.98 | LT05_L1TP_194028_20070222_20180118_01_T1_sr_band2 | 5,500 | 106 (SRTM-1, GLO-90), 107 (swissALTI3D) |
| Great Aletsch | 18.02.2009 | TM | 2 | 151.65 | 28.13 | LT05_L1TP_195028_20090218_20180302_01_T2_sr_band2 | 5,500 | 109 (SRTM-1, swissALTI3D), 108 (GLO-90) |

| | | | | | | | | |
|---|---|---|---|---|---|---|---|---|
| Great Aletsch | 21.02.2010 | TM | 2 | 152.59 | 29.46 | LT05_L1TP_195028_2010022 1_20161016_01_T1_sr_band2 | 5,500 | 108 (SRTM-1, swissALTI3D), 106 (GLO-90) |
| Great Aletsch | 08.02.2011 | TM | 2 | 153.85 | 25.08 | LT05_L1TP_195028_2011020 8_20161010_01_T1_sr_band2 | 5,500 | 109 (SRTM-1, GLO-90, swissALTI3D) |
| Great Aletsch | 25.02.2014 | OLI | 3 | 154.88 | 31.67 | LC08_L1TP_194028_2014022 5_20170425_01_T1_sr_band3 | 5,500 | 108 (SRTM-1, swissALTI3D), 107 (GLO-90) |
| Great Aletsch | 19.02.2015 | OLI | 3 | 155.22 | 29.38 | LC08_L1TP_195028_2015021 9_20170412_01_T1_sr_band3 | 5,500 | 107 (SRTM-1), 108 (GLO-90), 110 (swissALTI3D) |
| Great Aletsch | 15.02.2016 | OLI | 3 | 155.68 | 27.95 | LC08_L1TP_194028_2016021 5_20170329_01_T1_sr_band3 | 5,500 | 110 (SRTM-1), 109 (GLO-90), 111 (swissALTI3D) |
| Great Aletsch | 20.02.2018 | OLI | 3 | 165,56 | 21,33 | LC08_L1TP_194028_2018022 0_20180308_01_T1 | 5,500 | 92 (GLO-90) |
| Great Aletsch | 22.02.2019 | OLI | 3 | 155.72 | 27.70 | LC08_L1TP_195028_2019021 4_20190222_01_T1_sr_band3 | 5,500 | 113 (SRTM-1), 109 (GLO-90), 112 (swissALTI3D) |
| Baltoro | 11.02.1992 | TM | 3 | 141.60 | 30.55 | LT05_L1TP_148035_1992021 1_20170123_01_T1_sr_band2 | 5,500 | 42 |
| Baltoro | 22.02.1996 | TM | 3 | 135.05 | 31.43 | LT05_L1TP_148035_1996022 2_20170105_01_T1_sr_band2 | 5,000 | 38 |
| Baltoro | 24.01.2000 | ETM+ | 2 | 151.94 | 29.48 | LE07_L1TP_148035_2000012 4_20170213_01_T1_B2 | 200 | 31 |
| Baltoro | 15.02.2011 | TM | 2 | 147.36 | 34.91 | LT05_L1TP_148035_2011021 5_20161010_01_T1_sr_band2 | 5,000 | 30 |
| Baltoro | 10.02.2015 | OLI | 3 | 150.86 | 34.45 | LC08_L1TP_148035_2015021 0_20170413_01_T1_sr_band3 | 4,500 | 29 |
| Baltoro | 29.02.2016 | OLI | 3 | 147.82 | 40.67 | LC08_L1TP_148035_2016022 9_20170329_01_T1_sr_band3 | 5,000 | 16 |
| Baltoro | 02.02.2018 | OLI | 3 | 152.16 | 32.33 | LC08_L1TP_148035_2018020 2_20180220_01_T1_sr_band3 | 5,000 | 29 |
| Baltoro | 08.02.2020 | OLI | 3 | 151.29 | 33.83 | LC08_L1TP_148035_2020020 8_20200211_01_T1_sr_band3 | 5,000 | 28 |
| Gulkana | 13.03.1986 | TM | 2 | 157.84 | 21.25 | LT05_L2SP_068015_1986031 3_20200917_02_T1 | 3,300 (West) / 2,700 (East) | 42 (West) / 21 (East) |
| Gulkana | 30.03.1989 | TM | 2 | 157.45 | 29.41 | LT05_L2SP_067016_1989033 0_20200917_02_T1 | 3,300 / 3,000 | 20 / 10 |
| Gulkana | 19.03.1999 | TM | 2 | 160.07 | 25.25 | LT05_L2SP_066016_1999031 9_20200908_02_T1 | 3,900 / 3,800 | 21 / 8 |
| Gulkana | 14.03.2009 | TM | 2 | 162.21 | 23.80 | LT05_L1TP_066016_2009031 4_20160906_01_T1_sr_band2 | 5,500 | 22 / 8 |
| Gulkana | 04.03.2011 | TM | 2 | 162.93 | 19.74 | LT05_L2SP_066016_2011030 4_20200823_02_T1 | 2,900 / 2,700 | 24 / 22 |
| Gulkana | 06.03.2014 | OLI | 3 | 165.73 | 17.11 | LC08_L1TP_066016_2014022 4_20170306_01_T1_sr_band3 | 6,500 | 23 / 23 |

| | | | | | | | | |
|---|---|---|---|---|---|---|---|---|
| Gulkana | 27.02.2015 | OLI | 3 | 165.49 | 18.12 | LC08_L1TP_066016_2015022 7_20170227_01_T1_sr_band3 | 7,500 | 25 / 21 |
| Gulkana | 21.02.2016 | OLI | 3 | 165.60 | 15.81 | LC08_L1TP_067016_2016022 1_20170224_01_T1_sr_band3 | 7,000 | 25 / 23 |
| Gulkana | 02.03.2017 | OLI | 3 | 165.54 | 19.48 | LC08_L2SP_068016_2017030 2_20200905_02_T1 | 3,650 / 3,000 | 24 / 23 |
| Gulkana | 07.03.2018 | OLI | 3 | 165.56 | 21.33 | LC08_L2SP_066016_2018030 7_20200901_02_T1 | 4,000 / 3,500 | 25 / 20 |
| Gulkana | 22.02.2019 | OLI | 3 | 165.54 | 16.27 | LC08_L1TP_066016_2019022 2_20190308_01_T1_sr_band3 | 5,500 | 25 / 22 |
| Gulkana | 16.02.2020 and 25.02.2020 | OLI | 3 | 165.57 | 17.30 | LC08_L1TP_066016_2020022 5_20200313_01_T1_sr_band3 | 6,000 | 52 / 47 |
| Gulkana | 06.03.2021 | OLI | 3 | 165.62 | 21.05 | LC08_L2SP_067016_2021030 6_20210317_02_T1 | 4,400 / 3,000 | 25 / 23 |
| South Cascade | 02.02.1987 | TM | 2 | 148.87 | 18.90 | LT05_L1TP_046026_1987020 2_20161003_01_T1_sr_band2 | 5,000 | 34 |
| South Cascade | 18.02.1993 | TM | 2 | 147.04 | 23.98 | LT05_L1TP_046026_1993021 8_20160928_01_T1_sr_band2 | 5,000 | 32 |
| South Cascade | 05.02.1994 | TM | 2 | 148.49 | 19.79 | LT05_L1TP_046026_1994020 5_20160927_01_T1_sr_band2 | 5,000 | 33 |
| South Cascade | 11.02.1996 | TM | 2 | 144.09 | 20.01 | LT05_L1TP_046026_1996021 1_20160925_01_T1_sr_band2 | 5,000 | 29 |
| South Cascade | 22.02.1997 | TM | 2 | 147.53 | 25.73 | LT05_L1TP_045026_1997022 2_20160924_01_T1_sr_band2 | 5,000 | 31 |
| South Cascade | 29.01.2000 | ETM+ | 2 | 157.10 | 20.21 | LE07_L1TP_046026_2000012 9_20161003_01_T1_B2 | 100 | 32 |
| South Cascade | 20.02.2002 | TM | 2 | 150.97 | 25.98 | LT05_L1TP_045026_2002022 0_20160916_01_T1_sr_band2 | 5,000 | 32 |
| South Cascade | 07.02.2003 | TM | 2 | 151.18 | 21.32 | LT05_L1TP_045026_2003020 7_20160916_01_T2_sr_band2 | 5,000 | 33 |
| South Cascade | 10.02.2004 | TM | 2 | 152.29 | 22.59 | LT05_L1TP_045026_2004021 0_20160914_01_T2_sr_band2 | 5,000 | 33 |
| South Cascade | 15.02.2006 | TM | 2 | 154.72 | 25.25 | LT05_L1TP_045026_2006021 5_20160911_01_T1_sr_band2 | 5,000 | 29 |
| South Cascade | 02.02.2007 | TM | 2 | 157.05 | 21.44 | LT05_L1TP_045026_2007020 2_20160911_01_T1_sr_band2 | 5,000 | 31 |
| South Cascade | 07.02.2009 | TM | 2 | 154.31 | 22.46 | LT05_L1TP_045026_2009020 7_20160906_01_T1_sr_band2 | 5,000 | 32 |
| South Cascade | 17.02.2010 | TM | 2 | 154.56 | 25.93 | LT05_L1TP_046026_2010021 7_20160904_01_T1_sr_band2 | 5,000 | 32 |
| South Cascade | 20.02.2011 | TM | 2 | 154.07 | 26.82 | LT05_L1TP_046026_2011022 0_20160901_01_T1_sr_band2 | 5,000 | 33 |
| South Cascade | 15.02.2015 | OLI | 3 | 157.11 | 25.79 | LC08_L1TP_046026_2015021 5_20170301_01_T1_sr_band3 | 5,000 | 28 |

| South Cascade | 13.02.2017 | OLI | 3 | 157.28 | 25.30 | LC08_L1TP_045026_20170213_20180201_01_T2_sr_band3 | 5,000 | 31 |
|---|---|---|---|---|---|---|---|---|
| Sperry | 28.02.1986 | TM | 2 | 147.03 | 27.79 | LT05_L1TP_041026_19860228_20161004_01_T1_sr_band2 | 3,500 | 25 |
| Sperry | 19.02.2000 | TM | 2 | 149.88 | 25.04 | LT05_L1TP_041026_20000219_20160918_01_T1_sr_band2 | 3,500 | 11 |
| Sperry | 27.02.2003 | TM | 2 | 149.10 | 28.04 | LT05_L1TP_041026_20030227_20160916_01_T1_sr_band2 | 3,500 | 27 |
| Sperry | 19.02.2006 | TM | 2 | 154.37 | 26.61 | LT05_L1TP_041026_20060219_20160911_01_T1_sr_band2 | 3,500 | 12 |
| Sperry | 25.02.2008 | TM | 2 | 153.74 | 28.54 | LT05_L1TP_041026_20080225_20160906_01_T1_sr_band2 | 3,500 | 12 |
| Sperry | 25.02.2014 | OLI | 3 | 156.53 | 29.46 | LC08_L1TP_041026_20140225_20170307_01_T1_sr_band3 | 3,500 | 17 |
| Sperry | 28.02.2015 | OLI | 3 | 156.08 | 30.42 | LC08_L1TP_041026_20150228_20170301_01_T1_sr_band3 | 3,500 | 20 |


**Table A2: DEMs used to simulate shadows on glaciers, including spatial resolution, acquisition date, and data source.**

| DEM | Investigated Glacier | Spatial resolution [m] | Acquisition date | Source |
|---|---|---|---|---|
| swissALTI3D | Great Aletsch | (downsampled from 2 m to 5 m) | 2017-2018 | https://www.swisstopo.admin.ch/en/geodata/height/alti3d.html |
| ArcticDEM | Gulkana | 2 | 2009 | https://www.pgc.umn.edu/data/arcticdem SETSM_WV01_20090616_10200100079A2600_1020010007 D06000_seg1_2m_v3.0.tif (used item: SETSM_~1.TIF) |
| SRTM-1 | Great Aletsch, Baltoro, South Cascade, Sperry | ~30 (1-Arc second) | 2000 | http://www.opentopography.org |
| SRTM-3 | Great Aletsch, Baltoro | ~90 (3-Arc seconds) | 2000 | http://www.opentopography.org |
| NASADEM | Great Aletsch, Baltoro | 30 | 2000 | http://www.opentopography.org |
| ALOS World 3D (AW3D30) | Great Aletsch, Baltoro | 30 | 2006-2011 | http://www.opentopography.org |
| Copernicus Global DEM (GLO-30) | Great Aletsch, Baltoro | 30 | 2011-2015 | http://www.opentopography.org |
| Copernicus Global DEM (GLO-90) | Great Aletsch, Baltoro | 90 | 2011-2015 | http://www.opentopography.org |
| Viewfinderpanoramas DEM (VFP) | Baltoro | 90 | 2000 incl. void fill data with variable dates | http://viewfinderpanoramas.org/ |

**Table A3: Prior and posterior distributions of the parameters in the hierarchical models of glacier elevation change $\Delta h$ with year $y$ using bearing lines (Eqs. 1-11).**

| Parameter | Prior | Posterior<br>Median \| 2.5% \| 97.5% of HDI |
|:---:|:---:|:---:|
| $\alpha$ | Normal (mean = 0, sd = 2.5) | 0.23 \| -0.14 \| 0.61 |
| $\beta$ | Normal (mean = 0, sd = 2.5) | -0.31 \| -0.64 \| 0.03 |
| $\sigma_\alpha$ | Normal (mean = 0, sd = 2.5) T(0, ) | 0.43 \| 0.22 \| 0.90 |
| $\sigma_\beta$ | Normal (mean = 0, sd = 2.5) T(0, ) | 0.38 \| 0.19 \| 0.80 |
| $\kappa$ | Normal (mean = 0, sd = 2.5) T(0, ) | 0.55 \| 0.53 \| 0.57 |
| $\varsigma$ | LKJCholesky(1) on **R** | 0.74 \| -0.00 \| 0.98 |

Notes: Priors refer to standardised input data pairs of $\Delta h$ and $y$ using a mean of zero and unit standard deviation. T($\cdot$, $\cdot$) indicates a truncation of the distribution at a lower or upper boundary. sd, standard deviation. Degrees of freedom are constant ($v = 3$) and have no posterior estimate.

**Table A4: Prior and posterior distributions of the parameters in the hierarchial models of glacier elevation change $\Delta h$ with year $y$ using all data (within and outside the Landsat period) and only for data within the Landsat period, determined from reference DEMs and historical maps (Eqs. 1-11).**

| Parameter | Prior | Posterior for all available data<br>Median \| 2.5% \| 97.5% of HDI | Posterior for data from the Landsat era only<br>Median \| 2.5% \| 97.5% of HDI |
|:---:|:---:|:---:|:---:|
| $\alpha$ | Normal (mean = 0, sd = 2.5) | 0.17 \| -0.85 \| 1.16 | 0.17 \| -0.99 \| 1.32 |
| $\beta$ | Normal (mean = 0, sd = 2.5) | -0.48 \| -1.02 \| 0.06 | -0.32 \| -0.64 \| 0.04 |
| $\sigma_\alpha$ | Normal (mean = 0, sd = 2.5) T(0, ) | 1.03 \| 0.45 \| 2.42 | 1.19 \| 0.52 \| 2.69 |
| $\sigma_\beta$ | Normal (mean = 0, sd = 2.5) T(0, ) | 0.52 \| 0.45 \| 2.42 | 0.27 \| 0.01 \| 0.88 |
| $\kappa$ | Normal (mean = 0, sd = 2.5) T(0, ) | 0.26 \| 0.18 \| 0.37 | 0.21 \| 0.12 \| 0.34 |
| $\varsigma$ | LkjCholesky(1) on **R** | 0.44 \| -0.55 \| 0.97 | 0.01 \| -0.87 \| 0.88 |

Notes: Priors refer to standardised input data pairs of $\Delta h$ and $y$ using a mean of zero and unit standard deviation. T( $\cdot$, $\cdot$) indicates a truncation of the distribution at a lower or upper boundary. sd, standard deviation. Degrees of freedom are constant ($v = 3$) and have no posterior estimate.

**Table A5: Comparison of heights at stable terrain (ST) in Landeskarte over time and with different DEMS.**

| year | ST1 | ST2 | ST3 | ST4 | ST5 | ST6 | ST7 | ST8 | ST9 | ST10 |
|---|---|---|---|---|---|---|---|---|---|---|
| 1959 | 3465 | 3366 | 3242,6 | 4195 | 3810,7 | 3754 | 3641 (snow) | 2994,8 | 2951,7 | 3016,2 |
| 1968 | 3466 | 3366 | 3242,6 | 4195 | 3810,7 | 3754 | 3641 (snow) | 2994,8 | 2951,7 | 3016,2 |
| 1971 | 3466 | 3366 | 3242,6 | 4195 | 3810,7 | 3754 | 3641 (snow) | 2994,8 | 2951,7 | 3016,2 |
| 1975 | 3466 | 3366 | 3242,6 | 4195 | 3810,7 | 3754 | 3641 (snow) | 2994,8 | 2951,7 | 3016,2 |
| 1981 | 3463 | 3366 | 3242,6 | 4195 | 3810,7 | 3754 | 3639 | 2994,8 | 2951,8 | 3016,2 |
| 1987 | 3463 | 3366 | 3242,6 | 4195 | 3810,7 | 3754 | 3639 | 2994,8 | 2951,8 | 3016,2 |
| 1993 | 3463 | 3366 | 3242,6 | 4193 | 3810,7 | 3754 | 3639 | 2995 | 2952 | 3016 |
| 1999 | 3463 | 3366 | 3242,6 | 4193 | 3810,7 | 3754 | 3639 | 2995 | 2952 | 3016 |
| 2005 | 3463 | 3366 | 3242,6 | 4193 | 3810,7 | 3754 | 3639 | 2995 | 2952 | 3016 |
| 2011 | 3463 | 3366 | 3243 | 4193 | 3811 | 3754 | 3639 | 2995 | 2952 | 3016 |
| 2016 | 3463 | 3366 | 3243 | 4193 | 3811 | 3754 | 3639 | 2995 | 2952 | 3016 |
| 2020 | 3463 | 3366 | 3243 | 4194 | 3811 | 3756 | 3639 | 2995 | 2952 | 3016 |
| swissALTI3D | 3460,5 | 3364,4 | 3242,2 | NA | 3810,2 | 3754,95 | 3638,1 | 2995,98 | 2953,1 | 3020,9 |
| Cop90 | 3386,9 | 3277,9 | 3114,9 | 4133,2 | 3750,2 | 3694,3 | 3571,2 | 2858,1 | 2894,5 | 2956,98 |
| Cop30 | 3389,9 | 3314,8 | 3119,6 | 4144,4 | 3791,7 | 3702,8 | 3584,9 | 2903,5 | 2926,2 | 3003,4 |

**Figure A1: Webpage with historical maps (Landeskarte) from the Bundesamt für Landestopografie KOGIS (Koordination, Geoinformation und Services, https://www.swisstopo.admin.ch)**

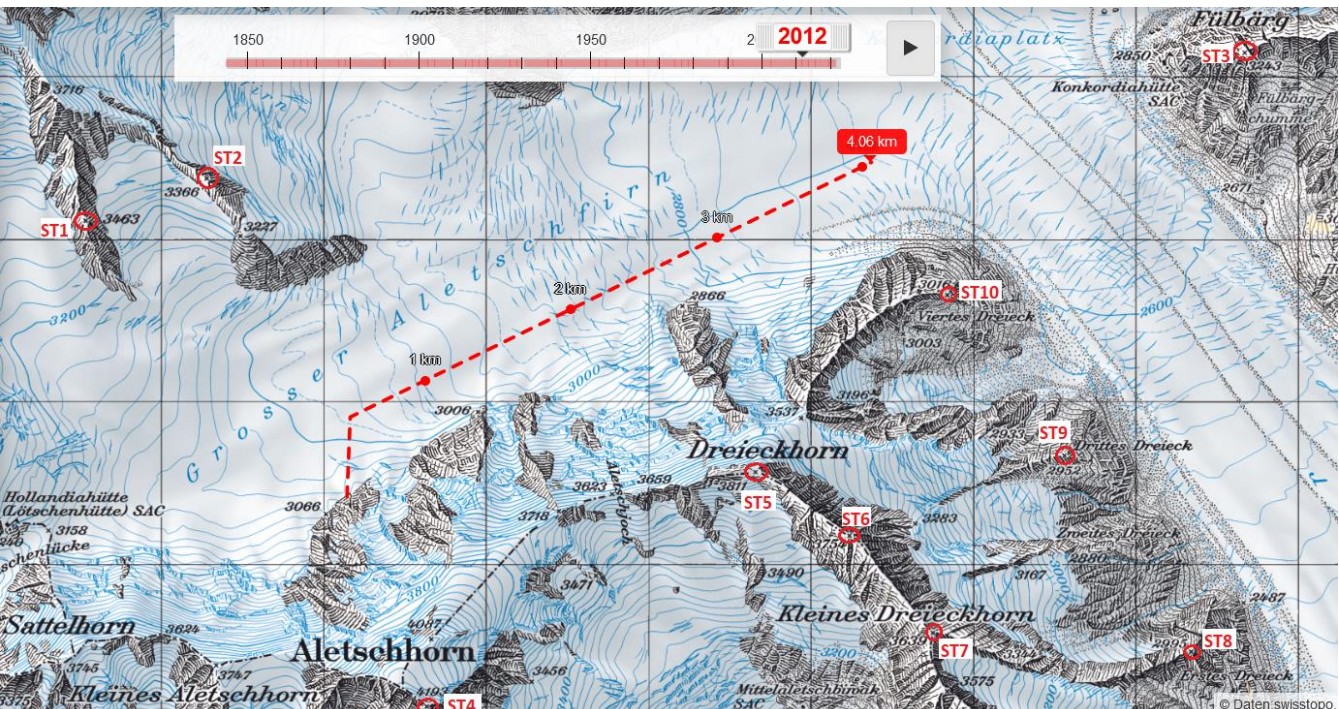

Notes: Red circles (ST1-ST10) represent locations of stable terrains (ST) that were investigated and compared with heights in different DEMs to proof the quality of the historical maps. Red dots represent locations we used to validate our results of glacier elevation changes.

## Data and code availability

The outlines of the shadows, the bearing lines, tables with inferred elevation changes for each glacier, and the Bayesian multi-level models are available via *Zenodo* (https://doi.org/10.5281/zenodo.8087360). Landsat images were obtained from *EarthExplorer* (https://usgs.earthexplorer.gov), and all DEMs from which we derived shadows are freely available from the sources provided in Table A2. DEMs for validation are available at https://alaska.usgs.gov/products/data/glaciers/benchmark_geodetic.php. Codes to fit the Bayesian multi-level models are available at *GitHub* (https://github.com/geveh/ShadowsOnGlaciers).

## Author contributions

All authors contributed equally in designing the study, conducting the analysis, validating the results, and writing the manuscript.

## Competing interests

The authors declare that they have no conflict of interest.

## Acknowledgements

We are indebted to the help from Romain Hugonnet who provided data on elevation changes along shaded glacier surfaces. This work is based on global DEM services provided by the OpenTopography Facility with support from the National Science Foundation under NSF Award Numbers 1948997, 1948994 & 1948857.

## Financial support

MP was supported for one year by a fellowship by the German Academic Exchange Service (DAAD) within the project Co-PREPARE, a collaboration between the University of Potsdam and the Indian Institute of Technology Roorkee.

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
