# Peer review of "Cast shadows reveal changes in glacier surface elevation"

_The Cryosphere, 2022_

## Referee Comment (RC1)

**Review The Cryosphere* – Cast shadows reveal changes in glacier thickness**
*Pfau et al.*

**General comments:**

This research proposes to estimate glacier surface elevation change by using the length of shadows cast by surrounding topography. Specifically, it relies on a reference DEM from which shadows are modelled at times corresponding to several Landsat acquisitions over the 1990-2020 period. From the imagery, a binary thresholding on the green band is used to map the actual shaded area. The proposed method then derives the change in glacier surface elevation along the boundary of cast shadows. This is done using the difference in length between the modelled and mapped shadows in the direction of illumination, and under the assumption of unchanged topographic gradient in that direction.

The method is tested on 5 glaciers that exhibit a prominent surrounding topography casting extensive shadows over parts of the glaciers. The SRTM 1'' DEM (acquisition Feb 2000) is used as reference DEM for 3 glaciers (Sperry, Aletsch, South Cascade); a variation of SRTM 3'' potentially mixed with other unknown data source (Viewfinder Panoramas DEM, VFP) is used for Baltoro, and ArticDEM for Gulkana. For each glacier and each landsat image, differences in shadow length are converted to height variations and analysed statistically with a Beyesian multi-level linear regression model to estimate linear trends of thickness change for each glacier. This suggests significant downwasting trends for Aletsch, Cascade and Sperry, while the author conclude thickening for Baltoro and no significant trend for Gulkana. A comparison of results with repeated DEMs is completed on all but Baltoro glacier. The effect of DEM source and resolution is assessed on Aletsch Glacier.

Overall, I find this contribution original and interesting but not overly convincing. It is clear and well written. The methodology is well and sufficiently explained, and the results can be reproduced. However, although limitations of the approach appear correctly identified, I find several shortcomings that require attention and significant revisions before this work can be considered for publication.

**Specific comments:**

I find the use of Viewfinder Panoramas DEM for Baltoro Glacier arguable. This DEM is of uncertain quality. The authors themselves state that "date of the map basis of VFP is not known". It also appears incorrectly referenced as 30m resolution in Table A2 although it is specified that VFP DEM in Asia is only at 3'' (http://viewfinderpanoramas.org/dem3.html). Figure 1 in this review compares the VFP DEM with the SRTM 1'' (30m) and CGIARSRTM v4.1 (90m) over Baltoro Glacier. It confirms the 3'' resolution of VFP. Figure 1(b) also shows that SRTM 1'' exhibits no hole that would compromise the shadow algorithm. Figure 1(d), however, demonstrates how different VFP is from SRTM 1'', in particular over areas of significance to render proper shadows. In view of this, I don't understand the choice made by the authors to mix VFP and SRTM. I believe the analysis of Baltoro should be redone on the basis of SRTM 1'' alone.

[Figure]

(a) VFP (N35E076.hgt, from http://www.viewfinderpanoramas.org/dem3/I43.zip, last retrieved 05/12/2022

(b) SRTM 1 Arc-Second Global (DOI:10.5066/F7PR7TFT) (tile n35_e076_1arc_v3.tif from USGS)

[Figure]

(c) CGIAR SRTM v4.1 (https://srtm.csi.cgiar.org/wp-content/uploads/files/srtm_5x5/TIFF/srtm_52_05.zip)

[Figure]

(d) VFP minus SRTM 1''

*Figure 1 Various DEM of Baltoro glacier and difference between VFP and SRTM 1''*

For Baltoro Glacier, the authors also state in P9L192 that no data are available for comparison. I would recommend that the authors give more consideration to Hugonnet et al. (2021) as data are readily available from https://www.theia-land.fr/en/monitoring-700000-km%C2%B2-of-the-worlds-glaciers/.

By curiosity, I plotted the 2000-2019 rate of surface elevation change from Hugonnet et al. (2021) for Baltoro Glacier (Figure 2a). The spatial variability in surface elevation change illustrates one major limitation of the proposed approach. It reveals how trends along a path that is limited to cast shadow can fail to resolve significant signal and trends for the rest of the glacier. The unambiguous negative trend visible from Hugonnet et al. (2021) also potentially contradicts results from this study (e.g., figure 4 and statement P10L218 "*Baltoro Glacier shows slight gains in glacier thickness*") which

cast concerns over the methodology and/or statistical testing. It may suggest that the inference derived from the statistical model are ill-informed or that the selective coverage of cast shadow is deceiving as it conceals the overall behaviour to the extent of drawing wrong conclusions. It appears necessary to revisit findings and conclusions with this in mind. Again, I am curious to see what would come from using the SRTM 1'' data as it may exemplify further the sensitivity of the method to the DEM.

[Figure]

[Figure]

(a) Baltoro, 2000-2019 (m/year)          (b) Gulkana, 2010-2019 (m/year)

*Figure 2 Surface elevation change form Hugonnet et al. 2021 ([https://www.theia-land.fr/en/product/rate-of-glacier-elevation-changes-from-2000-to-2019/](https://www.theia-land.fr/en/product/rate-of-glacier-elevation-changes-from-2000-to-2019/) )*

Another useful comparison can be made for Glkana Glacier. Hugonnet et al. (2021) map rates of change over the 2010-2019 period that are directly comparable with the trends and conclusion inferred by the authors from cast shadows. Figure 2(b) shows the contrasts in trends from the accumulation area with shadow cast by Ogive Mountain and those cast by Icefall Peak.

In this context, the authors state P10L217 that "*Annual rates of glacier elevation change at Gulkana Glacier are not credibly different from zero*", and strengthen their conclusion P11L237 by stating "*At Gulkana, both our method and high-resolution DEM suggest the highest uncertainties in the estimated trends, leaving little room for a credible trend in glacier elevation change*".

While I could conceive that the author's method finds not trend from shadow cast by Ogive Mountain as it would correspond to marginal rate of change in Figure 2(b), it would be expected that shadows cast by Icefall Peak yield a significantly negative signal. While revisiting the results in view of these data, it would be useful to separate signals from each mountain and compare critically with the rates assessed by Hugonnet et al. (2021). The conclusion that annual rate is not credibly different from zero must be reassessed as it either echoes again a significant limitation of the method, or it compromises findings from Hugonnet et al. (2021). At this stage and with the evidence provided by the authors, I believe the former remains more credible.

By contrast, rate inferred for Sperry, South Cascade, and Aletsch seem to compare better with Hugonnet et al. (2021) although the trends derived over the 1990-2020 period may subdue that assessed by Hugonnet et al. (2021) over the 2000-2020 period. Such detail assessment with a consistent dataset would be desirable and will provide more perspective on the validity and limitations of the proposed approach, while also shedding light on the contrasted and generally

unconvincing agreements found by the authors with trends derived from repeated DEMs and historical maps.

Finally, the authors assess the variability of shadow predicted over Aletsch Glacier from various DEMs. This is a useful and well-thought comparison that does inform about uncertainties associated with resolving shadows. Nonetheless, I find the assessment falls short of considering the effect of using these different DEMs on determining a trend of elevation change. It would be necessary that the authors repeat the full analysis on Aletsch with each DEMs to fully determine how DEM propagate uncertainties into the linear model.

**Technical corrections**

P6L127 : "manually mapped shadow" should better be called "shadow derived from Landsat images" as it is not mapped manually but rather derived via thresholding on the Green band.

P6L129: A *geodetic line* is defined as the shortest distance between two points on the surface of the ellipsoid. I don't think this is a relevant name for what is used here, namely a set of regularly spaced line in the direction of the sun at the time of image acquisition.

P19&20: Notes in both Tables A3 and A4 should read "at a lower" instead of "at an lower".

P19TableA2: ViewFinder Panoramas DEM of Baltoro is ~90m (3'' for ASIA, see http://viewfinderpanoramas.org/dem3.html).

P19TableA2: There is no SRTM 1'' for Gulkana glacier

---

## Referee Comment (RC2)

**Review: Pfau et al., Cast shadows reveal changes in glacier thickness (tc-2022-194)**

The authors of the manuscript present a promising new way of extracting glacier elevation change from the changing extent of shadows cast from direct sunlight. The manuscript clearly portrays the potential of this approach, and important discussion is made on the potential of scaling this approach to regional or global scope. I find the validation analyses slightly unconvincing, however; a general agreement between this approach and one "reference" classical DEM comparison of questionable quality is found at Grosser Aletschgletscher, which is used as the main argument for the use of this approach. From a critical point of view, one general agreement is not enough to prove the potential of this method. If the authors either had another convincing case-study, or put less emphasis on this one agreement in the text, it would not be as large of an issue. I also do not consider absolute proof necessary for this study to be a novel contribution and an interesting read. I have comments on the text and some details of the analysis, but I generally find it well written and not in need of significant revision. I therefore suggest minor revisions of this work, and I look forward to be able to share this with my colleagues when published in its revised state!

**General comments**

In the end of the discussion, you mention the potential to automate and therefore efficiently scale the analysis. This is what I think is the biggest take-home message, and I suggest that it should be reflected in the abstract. Doing this manually on single glaciers is only so impactful, but the fact that this approach utilises globally available data, with relatively small automation challenges, speaks to the massive potential of this approach!

I was initially worried about the use of terrain-corrected Landsat images, but with some handkerchief mathematics, I could calm myself down. I suggest modifying the reasoning that I present here and putting it in writing, as it strenghens your choice of data. My worry was that the images will be skewed depending on what DEM is used by USGS to correct them. Optimally, one should run the correction with the same DEM as the one you use for shadow simulation, which would of course add significant computational overhead to the approach. If that is not done, there will be an error that is dependent on the DEM difference and the incidence angle (the relative angle between the pixel coordinate and the satellite at the time of acquisition). Landsat 8 orbits at an altitude of 705 km and its OLI instrument has a swath width of 185 km according to Wikipedia, meaning that the incidence angle varies between 0 and about 7.5 degrees. This means that the DEM bias that is required to shift the image by one pixel is around 230 m, which is extremely unlikely. Therefore, it should not matter too much, meaning that a potentially incorrect terrain correction is fine. If this is brought up in the text, it would save others having to go down the same rabbit hole as me!

References are sorted inconsistently in the text. Sometimes they are sorted by year and sometimes alphabetically. This should be made consistent in accordance to the journal standard (which I believe is by year).

This comment requires no changes to the manuscript; it is simply a suggestion for the future. Filtering of the geodetic lines as shown in Figure 3e could easily be done programmatically. If

all lines are given an index before cutting them, then the cut line that is furthest away from the sun for each index would be the line to keep, whereas all others are erroneous.

I am not a Bayesian wizard, and therefore needed help from my colleagues to understand the statistics that are perfomed in this manuscript. I was told that this is a good approach as it is simple and understandable for anyone that is into the field. For the others (like me), however, could you keep that in mind for Section 3.2? For example, it is not argued for why this is preferable over a simpler optimization method (e.g. least squares). I now understand that this approach simply keeps the outgoing uncertainty high when provided with few data, whereas a regular regression would result in one misleading line that may be completely wrong. An easy argument for your approach is that you have few data points, and therefore cannot use regular optimization tools. But this was not clear to me before I had spent a few hours figuring out what is going on. Also, in your equations, I see no factor for elevation. Elevation change is highly correlated with elevation (i.e hypsometric gradient), so different points at different elevation are expected to have different values. Can you argue in the text for why this is hopefully not the main reason for the spread you present later in the Results chapter? Because if it is, I would suggest you update the model accordingly.

Currently, there is no consideration of elevation change uncertainty between the DEM comparisons in Section 3.3. The scanned topographic map and the SwissALTI3D DEM of Grosser Aletschgletscher for example may have substantial offsets, which are usually quantified from neighbouring stable terrain. Since the North America DEMs have gone through peer-review, they must have an associated uncertainty in the publication. For Grosser Aletschgletscher, I suggest validating that differences on stable terrain are close enough to zero and putting it in writing, or in a new figure. Without considering this source of error, I would not trust the validation from Switzerland, and therefore cannot trust the statement of similar trends on L230.

Please validate the date of the ArcticDEM product you used over Gulkana Glacier. The ArcticDEM explorer only shows strips from 2011–2016 in that location, while you claim that your DEM is from 2009.

**Specific comments**

- L18: Mention that the thinning pertains to the points of measurement, not the entire glacier. Otherwise, people might cite this number incorrectly.

- L36: Is 141,000 km³ for mountain glaciers or total land ice?

- L37: Change "cover" to "covers"; magnitudes are in singular.

- L40: Change "elevations" to "elevation"; magnitudes are in singular.

- L41: Change "(ICESat)" to "(e.g. ICESat)".

- L44: There are more large-scale long-term studies (e.g. Belart et al., 2020; Geyman et al., 2022; Mannerfelt et al., 2022).

- L48: All Landsat programme products do not have a resolution of 30 m. Only most bands of the TM, ETM+ and OLI instruments (see the Landsat program wikipedia article).

- L49: Change "mapped in Landsat" to "mapped from Landsat"

- L51: Change "help reveal" to "be used to estimate" or similar. Outlines themselves cannot reveal elevation change. The first reference uses area for a rough estimation of volume change, and the second uses them to crop their DEM differencing.

- L64: The DEM abbreviation is already introduced in the abstract, and the DEM abbreviation is used in L49.

- L81: Change "long(decadal)" to "decadal"

- L86: The line starting with "High and steep" is very short, and could be rephrased or combined with another.

- L87: Specify that "Concordia" is the name of a mountain.

- L89: Change "Gulkana" to "Gulkana glacier" to be consistent with L90 and to help the reader. I know that the word "glacier" occurs in the beginning of the sentence, but the wording is still not obvious to me.

- L105: Did you validate the georeferencing? Sometimes they can be off if not enough GCPs exist. Please just mention that this was (hopefully) not a problem.

- L109: Which ArcticDEM product was used? The mosaic or an individual strip? If an individual strip, which id?

- L112: Do you mean the swissALT3D DEM 2019 version? The acquisition year over Grosser Aletschgletscher is 2017–2018 as is correctly mentioned later.

- L122: It would be nice to know a bit more about why the green band was chosen. For example, did it have the highest difference between shaded and unshaded locations? It is mentioned that "shadows appear dark" in the green band, but not "darker" than the other bands.

- L125: Does the algorithm in SAGA account for the Earth's curvature? The effect is in the order of 1 m in elevation per 3 km of horizontal distance, so this would matter somewhat at these scales. Please just mention if it is already accounted for, or mention that future improvements can be made by accounting for it.

- L128: Specify that it's RGI version 6.

- L187: Are you entirely sure that the maps represent these exact years? Swiss maps are very often asynchronously updated over time, and can have large (tens of metres) differences to what one would expect (c.f. Fig. 10 in Mannerfelt et al., 2022).

- L207: The swissALTI3D DEM is a mix of lidar at low elevation and aerial photogrammetry at high elevation (see the technical information on swisstopo's website). At Grosser Aletschgletscher, it is therefore most likely based on photogrammetry.

- L211: Repeat what the line spacing is. 45 lines is not informative unless the spacing is mentioned.

- L231: The topographic maps may have a high resolution, but it is far from guaranteed to have a high accurancy and precision. I would rephrase this from "high-resolution DEMs" to "reference DEM comparison".

- L244: Is the substantial variance found before or after considering the elevation dependent elevation change that is observed on almost every glacier in the world? If they are at relatively similar elevation, then say that: "... in spite of their similar elevation ...".

- L252: Where is the date of the GLO-90 DEM information from? On the associated OpenTopography website it says 2011–2015

- L277: Right now, you only show a difference in the horizontal variability that is associated with different DEMs, not the effect on calculated elevation change. My Figure 7 suggestion below would solve this problem.

- L294: The effects of SAR penetration would lead to a potential negative bias (longer shadows), no? But since this bias will be consistent between years, one would just have to subtract the shadow-derived elevation change at the year of acquisition of the DEM. So this is arguably a very simple fix. Could this be put in words, assuming you agree with me?

- L302: "Precision" (spread) AND "accuracy" (bias). There seems to be both consistent and inconsistent errors in the shadow-maps derived from poor DEMs.

- L309: Please elaborate on why you think winter months are better suited for this method. Lower solar angles, I presume? What about the contrast between shaded and unshaded terrain?

- L330: Steep topography is arguably not a requirement; just stable topography. Indeed, steep slopes increase the potential time of day at which shadows can be created, so it works better with steeper topography. But the approach still works in shallow topography; just at fewer hours of the day! On e.g. ice caps, there is no stable topography to cast shadows, so the approach fails. So steep topography and high latitudes are preferable, but stable topography is the only theoretical requirement.

- Figure 1: I just want to add that I think this is a great figure that explains the concept simply and artfully! Great job!

- Figure 2: The Randolph Glacier Inventory is mentioned without abbreviating it, contrasting it to Figure 3.

- Figure 3: The RGI abbreviation is introduced multiple times. Only RGI is necessary here.

- Figure 5: It is unclear where the grey bubbles come from. Are these from other satellite images than Landsat? Please clarify this in the text.

- Figure 7: I recommend to add another axis label on top where you show calculated height change assuming the solar parameters you decided. This would solve my issue in L277 that you have actually not shown the variable DEM quality effect on elevation change.

**References**

Belart JMC, Magnússon E, Berthier E, Gunnlaugsson AT, Pálsson F, Aðalgeirsdóttir G, Jóhannesson T, Thorsteinsson T, Björnsson H. 2020. Mass Balance of 14 Icelandic Glaciers, 1945–2017: Spatial Variations and Links With Climate. Frontiers in Earth Science 8:163. doi: 10.3389/feart.2020.00163.

Geyman EC, van Pelt WJJ, Maloof AC, Faste Aas H, Kohler J. 2022. Historical glacier change on Svalbard predicts doubling of mass loss by 2100. Nature 601:374–395. doi:10.1038/s41586-021-04314-4.

Mannerfelt ES, Dehecq A, Hugonnet R, Hodel E, Huss M, Bauder A, Farinotti D. 2022. Halving of Swiss glacier volume since 1931 observed from terrestrial image photogrammetry. The Cryosphere Discussions :1–32doi:10.5194/tc-2022-14.

---

## Author Comment (AC1)

**Anonymous Reviewer #1**

We thank the reviewer for the constructive comments. In our letter, we highlight the comments from the reviewer in orange. Our responses are in black font, and our planned corrections are highlighted in bold font.

**General comments:**

**R1C1:** This research proposes to estimate glacier surface elevation change by using the length of shadows cast by surrounding topography. Specifically, it relies on a reference DEM from which shadows are modelled at times corresponding to several Landsat acquisitions over the 1990-2020 period. From the imagery, a binary thresholding on the green band is used to map the actual shaded area. The proposed method then derives the change in glacier surface elevation along the boundary of cast shadows. This is done using the difference in length between the modelled and mapped shadows in the direction of illumination, and under the assumption of unchanged topographic gradient in that direction.

The method is tested on 5 glaciers that exhibit a prominent surrounding topography casting extensive shadows over parts of the glaciers. The SRTM 1'' DEM (acquisition Feb 2000) is used as reference DEM for 3 glaciers (Sperry, Aletsch, South Cascade); a variation of SRTM 3'' potentially mixed with other unknown data source (Viewfinder Panoramas DEM, VFP) is used for Baltoro, and ArcticDEM for Gulkana. For each glacier and each Landsat image, differences in shadow length are converted to height variations and analysed statistically with a Bayesian multi-level linear regression model to estimate linear trends of thickness change for each glacier. This suggests significant downwasting trends for Aletsch, Cascade and Sperry, while the author conclude thickening for Baltoro and no significant trend for Gulkana. A comparison of results with repeated DEMs is completed on all but Baltoro glacier. The effect of DEM source and resolution is assessed on Aletsch Glacier.

Overall, I find this contribution original and interesting but not overly convincing. It is clear and well written. The methodology is well and sufficiently explained, and the results can be reproduced. However, although limitations of the approach appear correctly identified, I find several shortcomings that require attention and significant revisions before this work can be considered for publication.

**R1A1:** We thank the reviewer for this positive feedback on our work. In our responses below, we address any of their concerns and detail how we will solve the issues in a revised manuscript. In summary, we will emphasize that our trends in glacier elevation change are valid only for areas on glaciers covered with shadows from adjacent mountains. We will further assess the impact of different DEMs on the simulated shadows and associated trends of glacier elevation change. Finally, our revised manuscript will also consider the data on glacier elevation change from Hugonnet et al. (2021) and thus offer a stringent comparison to one of the most elaborate datasets of its kind to date.

**Specific comments:**

**R1C2:** I find the use of Viewfinder Panoramas DEM for Baltoro Glacier arguable. This DEM is of uncertain quality. The authors themselves state that "date of the map basis of VFP is not known". It also appears incorrectly referenced as 30m resolution in Table A2 although it is specified that VFP DEM in Asia is only at 3'' (http://viewfinderpanoramas.org/dem3.html). **Figure R1C1** in this review compares the VFP DEM with the SRTM 1'' (30m) and CGIARSRTM v4.1 (90m) over Baltoro Glacier. It confirms the 3'' resolution of VFP. **Figure R1C1b** also shows that SRTM 1'' exhibits no hole that would compromise the shadow algorithm. **Figure R1C1d**, however, demonstrates how different VFP is from SRTM 1'', in particular over areas of significance to render proper shadows. In view of this, I don't understand the choice made by the authors to mix VFP and SRTM. I believe the analysis of Baltoro should be redone on the basis of SRTM 1'' alone.

[Figure]

(a) VFP (N35E076.hgt, from http://www.viewfinderpanoramas.org/dem3/I43.zip, last retrieved 05/12/2022)

(b) SRTM 1 Arc-Second Global (DOI:10.5066/F7PR7TFT) (tile n35_e076_1arc_v3.tif from USGS)

(c) CGIAR SRTM v4.1 (https://srtm.csi.cgiar.org/wp-content/uploads/files/srtm_5x5/TIFF/srtm_52_05.zip)

(d) VFP minus SRTM 1''

**Figure R1C1:** Various DEMs of Baltoro glacier and difference between VFP and SRTM 1''.

**R1A2:** The reviewer is right that the VFP-DEM has a resolution of 3''. **We will change this statement to "90 m (3 arc seconds)" in Table A2 accordingly.**

We initially used SRTM 1'' data (doi: 10.5066/F7PR7TFT; shown by the reviewer in **Figure R1C1b**) to cast shadows from Mitre Peak on the surface of Baltoro Glacier. Yet the accuracy of SRTM DEM decreases in the Higher Himalayas as elevation and steepness increase. In addition, the SRTM features regions with missing data (voids) (Mukul et al. 2017, Liu et al., 2019). In **Figure R1C1b**, the reviewer used a void-filled derivate of the original SRTM data

according to the online documentation: "SRTM 1 Arc-Second Global (Digital Object Identifier (DOI) number: /10.5066/F7PR7TFT) elevation data offer worldwide coverage of void filled data at a resolution of 1 arc-second (30 meters)". **Figure R1A1** confirms that Mitre Peak is void-filled according to the non-void filled original product (doi: 10.5066/F7K072R7). These voids have been filled "*using interpolation algorithms in conjunction with other sources of elevation data*" (https://www.usgs.gov/centers/eros/science/usgs-eros-archive-digital-elevation-shuttle-radar-topography-mission-srtm-non). It remains unknown which method or data USGS EROS used to approximate the elevation of Mitre Peak in SRTM 1''.

[Figure]

**Figure R1A1:** Voids (green) in SRTM 3'' data on mountains adjacent to Baltoro glacier. Mitre Peak is within a void, suggesting that its elevation in SRTM 1'' data was estimated using either interpolation or unspecified data by USGS EROS.

This uncertainty motivated us to fill the void in Mitre Peak in SRTM 3'' with data from View Finder Panoramas (VFP). VFP has higher accuracy in steep terrain than SRTM 3'' (see Fig. 6 in Liu et al. 2019). We recall that we left the SRTM 3'' data for the flat surface of Baltoro Glacier unchanged. Thus, we provide a seamless DEM of VPF for Mitre Peak and SRTM 3'' for Baltoro Glacier. It is the choice of the interpolation algorithm or ancillary data that explains the difference of ~100 m between VPF and SRTM 1'' in **Figure R1C1d** provided by the reviewer.

In any case, the underlying data source for Mitre Peak remains unknown in either data set. The reviewer thus raised the important question as to which the choice of the DEM will change the shape and area of shadows casted from steep mountain peaks. Decreasing the grid resolution of DEMs (i.e. increasing the cell size) acts as a low-pass filter on the topography, degrading features such as sharp ridgelines, narrow valley bottoms, and local topographic roughness generated by bedrock outcrops (Gao 1997, Grieve et al. 2016). DEMs of higher resolution (i.e. smaller cell size) might better preserve the distinct shape of mountains.

To assess the impact of DEM resolution on cast shadows, we compared the elevation in the VFP DEM with reported values in the literature and other globally available DEMs (**Figure R1A2**). Accordingly, the elevation of Mitre Peak from VFP (6066 m) is most consistent with reported values of its elevation ranging between 6010 m (https://en.wikipedia.org/wiki/Mitre_Peak,_Pakistan) and 6030 m (https://www.himalaya-info.org/Map%20karakorum_baltoro.htm). The vertical datum of the reported elevations remains unknown, but differences of ~23 m in elevation between the WGS 84 ellipsoid and the

EGM96 geoid (the vertical datum of the SRTM and VFP) can largely account for this offset. The other DEMs feature consistently lower elevations for Mitre Peak and, generally, DEMs with 90 m resolution have lower peak elevations than their 30-m counterparts due to smoothing of high frequency signals. In comparing mapped to simulated shadows on Baltoro Glacier, we find that SRTM+VFP (lower left panel in **Figure R1A2)** closely approximated, and all other DEMs underestimated, the maximum elevation of Mitre Peak. We did not find any evidence for major rockfalls in high-resolution images, and thus assume that Mitre Peak in the VFP DEM is representative for its form in the year 2000, the acquisition date of the SRTM.

[Figure]

[Figure]

[Figure]

[Figure]

[Figure]

[Figure]

[Figure]

**Figure R1A2:** Shadows of the Mitre Peak derived from different DEMs. Elevations provided for each panel refer to the elevation of Mitre Peak obtained from the different DEMs. The red outline shows the shadow for year 2000 with an azimuth angle of 151.94° and a sun elevation of 29.48°.

**R1C3:** For Baltoro Glacier, the authors also state in P9L192 that no data are available for comparison. I would recommend that the authors give more consideration to Hugonnet et al. (2021) as data are readily available from https://www.theia-land.fr/en/monitoring-700000-km%C2%B2-of-the-worlds-glaciers/.

**R1A3:** We thank the reviewer for bringing this study to our attention. Hugonnet et al. (2021) produced time series of automatically generated DEMs from the Advanced Spaceborne Thermal Emission and Reflection Radiometer (ASTER) satellite mission between 2000 and 2019. Similar to our assessment, Hugonnet et al. (2021) estimated cumulative and mean rates of glacier elevation change in this period from a number of DEMs per glacier using stereo-photogrammetry. Data on mean rate of glacier elevation change are available either as rasters at a cell resolution of 100 m × 100 m, or as tables showing trends for the entire glacier area between 2000 and 2019 (https://doi.org/10.6096/13). Neither of the products can therefore be used directly to quantify the change in glacier elevation in the shaded areas at the same time points as in our analysis. Yet, we are pleased that the lead author Romain Hugonnet has kindly agreed to extract the entire time series of glacier elevation changes for the shaded areas of the glaciers only. **In our revised manuscript, we will therefore include a new figure that**

**compares the rates of glacier elevation change within shadows between this study and ours, and discuss any related inconsistencies.**

**R1C4:** By curiosity, I plotted the 2000-2019 rate of surface elevation change from Hugonnet et al. (2021) for Baltoro Glacier (**Figure R1C4a**). The spatial variability in surface elevation change illustrates one major limitation of the proposed approach. It reveals how trends along a path that is limited to cast shadow can fail to resolve significant signal and trends for the rest of the glacier. The unambiguous negative trend visible from Hugonnet et al. (2021) also potentially contradicts results from this study (e.g., figure 4 and statement P10L218 "*Baltoro Glacier shows slight gains in glacier thickness*") which cast concerns over the methodology and/or statistical testing. It may suggest that the inference derived from the statistical model are ill-informed or that the selective coverage of cast shadow is deceiving as it conceals the overall behaviour to the extent of drawing wrong conclusions. It appears necessary to revisit findings and conclusions with this in mind. Again, I am curious to see what would come from using the SRTM 1'' data as it may exemplify further the sensitivity of the method to the DEM.

[Figure]

(a) Baltoro, 2000-2019 (m/year)    (b) Gulkana, 2010-2019 (m/year)

**Figure R1C4:** Surface elevation change form Hugonnet et al. 2021 (https://www.theia-land.fr/en/product/rate-of-glacierelevation-changes-from-2000-to-2019/)

**R1A4:** We agree that our approach only informs about the changes in glacier surface elevation within the area of covered by shadows. We had repeatedly addressed this premise in our earlier manuscript (L18, L58, L60-61, L77, L144-145, L238, L275-276, 327-329,), and will further strengthen this concept in our revised version of the abstract: **"Accordingly, a shadow on Baltoro Glacier (Karakoram, Pakistan) suggests slight local increases in elevation between 1987 and 2020, while shadows on Great Aletsch Glacier (Switzerland) point to the most negative thinning rates of about 1 m per year. Our estimates of glacier elevation change are tied to the occurrence of mountain shadows, and may help complement field campaigns in regions that are difficult to access."** In the revised discussion, we will add: **"We stress that our results are tied to local changes of shadows casted from adjacent mountains. Thus, we caution against comparing our results with glacier-wide mass balances because these integrate over entire glaciers or elevation bands within glaciers, and may refer to different study periods. For example, Hugonnet**

**et al. (2021) estimate that the entire areas of Great Aletsch and South Cascade Glacier had elevation changes of -1.42 ± 0.1 and -0.66 ± 0.15 m yr$^{-1}$ (mean and 1σ error), respectively, in 2000-2019. Our estimates are lower (-1.08$^{+0.06}$/$_{-0.05}$ and -0.42$^{+0.11}$/$_{-0.11}$ m yr$^{-1}$, respectively) in the longer Landsat period, either because we measure elevation changes at higher parts of the glacier with possibly lower melt rates, or because glacier melt has accelerated in recent decades (Hugonnet et al. 2021).”** We will conclude our manuscript with: **“We demonstrate for four glaciers that our method provides quantitative information about local changes in glacier elevation over time that are consistent with independent DEMs of difference in shadow-covered areas.”**

In our revised manuscript, we will compare our trends in local glacier elevation change with those obtained by Hugonnet et al. (2021), see our reply **R1A3**. Yet we disagree that the trend at Baltoro is “unambiguously negative” at Baltoro. In **Figure R1C4**, the reviewer shows a map of mean annual rate of glacier elevation change in 2000-2019 provided by Hugonnet et al. (2021). We obtained the same data (**Figure R1A4a**), and also the error in elevation change (one standard deviation, **Figure R1A4b**). From both maps, we extracted all raster cells of the glacier area covered by the shadow from Mitre Peak in the period 2000-2019. In **Figure R1A4c**, we show both the mean and the error in glacier elevation change. We find that in each grid cell, the error is higher than the mean glacier elevation change, embracing both positive and negative trends. Thus, our findings are well within the uncertainties provided by Hugonnet et al. (2021) for the entire glacier, and possibly more accurate on a local scale.

[Figure]

**Figure R1A4:** Glacier elevation changes at Baltoro Glacier using gridded data (100 × 100 m) from Hugonnet et al. (2021). **a,** Mean glacier elevation change (dhdt) and **b,** Uncertainty in glacier elevation change (err_dhdt) in the period 2000-2019. We extracted value pairs of dhdt and err_dhdt from the shadow (black outline) cast on Baltoro glacier. **c,** dhdt and err_dhdt for each pixel in the shadow. While trends are largely negative, the errors allow for positive values within the shaded area, consistent with our results.

**R1C5:** Another useful comparison can be made for Gulkana Glacier. Hugonnet et al. (2021) map rates of change over the 2010-2019 period that are directly comparable with the trends and conclusion inferred by the authors from cast shadows. **Figure R1C4 (b)** shows the contrasts in trends from the accumulation area with shadow cast by Ogive Mountain and those cast by Icefall Peak. In this context, the authors state P10L217 that “*Annual rates of glacier elevation change at Gulkana Glacier are not credibly different from zero*”, and strengthen their conclusion P11L237 by stating “*At Gulkana, both our method and high-resolution DEM suggest the highest uncertainties in the estimated trends, leaving little room for a credible trend in glacier elevation change*”.

While I could conceive that the author's method finds not trend from shadow cast by Ogive Mountain as it would correspond to marginal rate of change in **Figure R1C4b**, it would be expected that shadows cast by Icefall Peak yield a significantly negative signal. While revisiting the results in view of these data, it would be useful to separate signals from each mountain and compare critically with the rates assessed by Hugonnet et al. (2021). The conclusion that annual rate is not credibly different from zero must be reassessed as it either echoes again a significant limitation of the method, or it compromises findings from Hugonnet et al. (2021). At this stage and with the evidence provided by the authors, I believe the former remains more credible.

**R1A5:** We agree that these shadows need to be treated separately because they are cast at different elevations on Gulkana glacier, i.e., at ~1750m m for the shadow from Icefall Peak and at ~1800m m for the shadow from Ogive Mountain. **In our revised manuscript, we will calculate the trends in glacier elevation changes for both shadows separately and revise all figures and statements accordingly. This revision will also allow us to discuss how robustly our method can detect glacier elevation changes at different elevation bands along a glacier. The discussion of this analysis will also refer to the data trimmed to the shadow area, which we will receive soon from Romain Hugonnet.**

**R1A6:** By contrast, rate inferred for Sperry, South Cascade, and Aletsch seem to compare better with Hugonnet et al. (2021) although the trends derived over the 1990-2020 period may subdue that assessed by Hugonnet et al. (2021) over the 2000-2020 period. Such detail assessment with a consistent dataset would be desirable and will provide more perspective on the validity and limitations of the proposed approach, while also shedding light on the contrasted and generally unconvincing agreements found by the authors with trends derived from repeated DEMs and historical maps.

**R1A6: In our revised manuscript, we will calculate trends in glacier elevation change constrained to the same period (2000-2019) as in Hugonnet et al. (2021). We will add a table that compares our trends with data from Hugonnet et al. (2021) both for the entire glacier and the area trimmed to the shadow.**

**R1A7:** Finally, the authors assess the variability of shadow predicted over Aletsch Glacier from various DEMs. This is a useful and well-thought comparison that does inform about uncertainties associated with resolving shadows. Nonetheless, I find the assessment falls short of considering the effect of using these different DEMs on determining a trend of elevation change. It would be necessary that the authors repeat the full analysis on Aletsch with each DEMs to fully determine how DEM propagate uncertainties into the linear model.

**R1A7:** We agree that a comparison of different DEMs will help quantifying the impact of the underlying DEM in our workflow. **To this end, we will select three input DEMs, swissALTI3D, SRTM, and COP90 DEM, to cover the entire range of available raster resolutions, i.e. 5, 30, and 90 m, in our analysis. We will then repeat the steps to calculate the difference between modelled and Landsat-derived shadows to see how the trends in glacier elevation change vary based on the underlying DEM.**

**Technical corrections**

**R1C8:** P6L127: "manually mapped shadow" should better be called "shadow derived from Landsat images" as it is not mapped manually but rather derived via thresholding on the Green band.

**R1A8: We will correct this statement accordingly.**

**R1C9:** P6L129: A *geodetic line* is defined as the shortest distance between two points on the surface of the ellipsoid. I don't think this is a relevant name for what is used here, namely a set of regularly spaced line in the direction of the sun at the time of image acquisition.

**R1A9: We will replace "geodetic line" with "bearing line" throughout the manuscript.**

**R1C10:** P19&20: Notes in both Tables A3 and A4 should read "at a lower" instead of "at an lower".

**R1A10: We will change our wording accordingly.**

**R1C11:** P19TableA2: ViewFinder Panoramas DEM of Baltoro is ~90m (3'' for ASIA, see http://viewfinderpanoramas.org/dem3.html).

**R1A11: We will correct the resolution of VFP in Table A2 accordingly.**

**R1C12:** P19TableA2: There is no SRTM 1'' for Gulkana glacier

**R1A12: We will delete "Gulkana" in this cell accordingly.**

**REFERENCES**

Earth Resources Observation And Science (EROS) Center (2018a): Shuttle Radar Topography Mission (SRTM) Non-Void Filled. DOI: 10.5066/F7K072R7.

Earth Resources Observation And Science (EROS) Center (2018b): USGS EROS Archive - Digital Elevation - Shuttle Radar Topography Mission (SRTM) 1 Arc-Second Global. Available online at https://www.usgs.gov/centers/eros/science/usgs-eros-archive-digital-elevation-shuttle-radar-topography-mission-srtm-1?qt-science_center_objects=0#qt-science_center_objects, updated on 2/13/2023, checked on 2/13/2023.

GAO, J. A.Y. (1997): Resolution and accuracy of terrain representation by grid DEMs at a micro-scale. In *International Journal of Geographical Information Science* 11 (2), pp. 199–212. DOI: 10.1080/136588197242464.

Grieve, Stuart W. D.; Mudd, Simon M.; Milodowski, David T.; Clubb, Fiona J.; Furbish, David J. (2016): How does grid-resolution modulate the topographic expression of geomorphic processes? In *Earth Surf. Dynam.* 4 (3), pp. 627–653. DOI: 10.5194/esurf-4-627-2016.

Hugonnet, Romain; McNabb, Robert; Berthier, Etienne; Menounos, Brian; Nuth, Christopher; Girod, Luc et al. (2021a): Accelerated global glacier mass loss in the early twenty-first century. In *Nature* 592 (7856), pp. 726–731. DOI: 10.1038/s41586-021-03436-z.

Hugonnet, Romain; McNabb, Robert; Berthier, Etienne; Menounos, Brian; Nuth, Christopher; Girod, Luc et al. (2021b): Accelerated global glacier mass loss in the early twenty-first century - Dataset. DOI: 10.6096/13.

Liu, Xiaodong; He, Pengcheng; Chen, Weizhu; Gao, Jianfeng (2019): Improving Multi-Task Deep Neural Networks via Knowledge Distillation for Natural Language Understanding. Available online at https://arxiv.org/pdf/1904.09482.

Mukul, Manas; Srivastava, Vinee; Jade, Sridevi; Mukul, Malay (2017): Uncertainties in the Shuttle Radar Topography Mission (SRTM) Heights: Insights from the Indian Himalaya and Peninsula. In *Sci Rep* 7 (1), p. 41672. DOI: 10.1038/srep41672.

Seyfferth, Guenter (2006): Die Berge des Himalaya. Available online at https://www.himalaya-info.org/Map%20karakorum_baltoro.htm, updated on 12/4/2022, checked on 2/13/2023.

Wikipedia (Ed.) (2017): Mitre Peak, Pakistan. Available online at https://en.wikipedia.org/w/index.php?title=Mitre_Peak,_Pakistan&oldid=798396406, updated on 1/9/2017, checked on 2/13/2023.

---

## Author Comment (AC2)

**Anonymous Reviewer #2**

We thank the reviewer for the constructive comments. In our letter, we highlight the comments from the reviewer in orange. Our responses are in black font, and our planned corrections are highlighted in bold font.

**R2C1:** The authors of the manuscript present a promising new way of extracting glacier elevation change from the changing extent of shadows cast from direct sunlight. The manuscript clearly portrays the potential of this approach, and important discussion is made on the potential of scaling this approach to regional or global scope. I find the validation analyses slightly unconvincing, however; a general agreement between this approach and one "reference" classical DEM comparison of questionable quality is found at Grosser Aletschgletscher, which is used as the main argument for the use of this approach. From a critical point of view, one general agreement is not enough to prove the potential of this method. If the authors either had another convincing case-study, or put less emphasis on this one agreement in the text, it would not be as large of an issue. I also do not consider absolute proof necessary for this study to be a novel contribution and an interesting read. I have comments on the text and some details of the analysis, but I generally find it well written and not in need of significant revision. I therefore suggest minor revisions of this work, and I look forward to be able to share this with my colleagues when published in its revised state!

**R2A1:** We thank the reviewer for emphasizing the value of our work and encouraging us to revise a few minor points. We see our study as a proof of concept, making researchers aware of this untapped resource to use shadows for quantifying local changes in glacier thickness. Our approach hinges on the availability of freely available data, which may bias our attention to glaciers with a high number of DEMs and satellite images. In our revised manuscript, we will study in detail the effect of DEM resolution at Baltoro Glacier (see our reply **R1A2** to Reviewer 1) and also compare our trends in glacier elevation change to those from Hugonnet et al. (2021). These revisions will allow for a better balance in representing the different glaciers and data sources in our study.

**General comments:**

**R2C2:** In the end of the discussion, you mention the potential to automate and therefore efficiently scale the analysis. This is what I think is the biggest take-home message, and I suggest that it should be reflected in the abstract. Doing this manually on single glaciers is only so impactful, but the fact that this approach utilises globally available data, with relatively small automation challenges, speaks to the massive potential of this approach!

**R2A2:** We thank the reviewer for this positive appraisal and recognition of our work. In our revised manuscript, we will stress the idea of the reviewer: **"We conclude that our approach has the potential to complement existing or future in situ measuring networks anywhere**

**on Earth where mountains shade parts of adjacent glaciers. We thus enrich glaciological and geodetic assessment with a new method that helps quantifying glacier elevation changes especially at high altitudes with limited access."**

**R2C3:** I was initially worried about the use of terrain-corrected Landsat images, but with some handkerchief mathematics, I could calm myself down. I suggest modifying the reasoning that I present here and putting it in writing, as it strengthens your choice of data. My worry was that the images will be skewed depending on what DEM is used by USGS to correct them.

Optimally, one should run the correction with the same DEM as the one you use for shadow simulation, which would of course add significant computational overhead to the approach. If that is not done, there will be an error that is dependent on the DEM difference and the incidence angle (the relative angle between the pixel coordinate and the satellite at the time of acquisition). Landsat 8 orbits at an altitude of 705 km and its OLI instrument has a swath width of 185 km according to Wikipedia, meaning that the incidence angle varies between 0 and about 7.5 degrees. This means that the DEM bias that is required to shift the image by one pixel is around 230 m, which is extremely unlikely. Therefore, it should not matter too much, meaning that a potentially incorrect terrain correction is fine. If this is brought up in the text, it would save others having to go down the same rabbit hole as me!

**R2A3:** Ideally, our workflow should make use of images that are all co-registered to a terrain-corrected master image. Co-registered images will increase the precision of our analysis and avoid systematic offsets. **In our revised manuscript, we will discuss the effects that may arise if images with more than a pixel offset will enter our workflow.** We agree with the reviewer that an additional source of uncertainty would be introduced if different DEMs were used for terrain correction for different scenes covering the region. However, according to the Landsat L1TP processing description (https://www.usgs.gov/landsat-missions/landsat-levels-processing), the elevation data used for relief displacement remain the same throughout the Landsat time series.

**R2C4:** References are sorted inconsistently in the text. Sometimes they are sorted by year and sometimes alphabetically. This should be made consistent in accordance to the journal standard (which I believe is by year).

**R2A4**: The journal's guidelines for authors suggest that "in terms of in-text citations, the order can be based on relevance, as well as chronological or alphabetical listing, depending on the author's preference". **We decided that we will sort in-text references by year in the revised manuscript.**

**R2C5:** This comment requires no changes to the manuscript; it is simply a suggestion for the future. Filtering of the geodetic lines as shown in Figure 3e could easily be done programmatically. If all lines are given an index before cutting them, then the cut line that is furthest away from the sun for each index would be the line to keep, whereas all others are erroneous.

**R2A5:** We thank the reviewer for this helpful suggestion. We consider implementing this idea when applying our approach to a larger scale.

**R2C6:** I am not a Bayesian wizard, and therefore needed help from my colleagues to understand the statistics that are performed in this manuscript. I was told that this is a good approach as it is simple and understandable for anyone that is into the field. For the others (like me), however, could you keep that in mind for Section 3.2? For example, it is not argued for why this is preferable over a simpler optimization method (e.g. least squares). I now understand that this approach simply keeps the outgoing uncertainty high when provided with few data, whereas a regular regression would result in one misleading line that may be completely wrong. An easy argument for your approach is that you have few data points, and therefore cannot use regular optimization tools. But this was not clear to me before I had spent a few hours figuring out what is going on.

**R2A6:** A key motivation for using multi-level models is that we can obtain parameters (e.g. intercepts and slopes) for elevation change of different glaciers within one single model. Thus we can learn both the parameters for all glaciers and the variation between parameters by using a set of individual glaciers that are part of a larger population of glaciers. Multi-level models are advantageous for datasets with a different number of observations in each group, or in our case glacier, in which one glacier might have hundreds of bearing lines (e.g. Great Aletsch Glacier) and others might have fewer data (e.g. Gulkana Glacier). The hierarchical model structure avoids over-fitting parameters for glaciers with many bearing lines and generally improves inference for groups with few data points.

Bayesian inference estimates all model parameters based on the data (the likelihood) and prior information about those parameters. If a glacier had few bearing lines in some years, the posterior distribution of parameter estimates will be broad and vice versa. The annual trends in glacier elevation change depend entirely on the available data, and the posterior distributions reflect the uncertainties accordingly. In essence, our model could also estimate the trends for two or one or even no years of mapped bearing lines, a situation that is difficult to deal with in ordinary least squares regression. This estimate would correspond to the mean trend derived from all glacier shadows included in our analysis. The more bearing lines we map for a new, unobserved year at a particular glacier, the more specific our estimated trends will be for that glacier if its trend deviates from the global mean of the entire population in our dataset.

**R2C7:** Also, in your equations, I see no factor for elevation. Elevation change is highly correlated with elevation (i.e hypsometric gradient), so different points at different elevation are expected to have different values. Can you argue in the text for why this is hopefully not the main reason for the spread you present later in the Results chapter? Because if it is, I would suggest you update the model accordingly.

**R2A7**: We are unsure how to interpret the reviewer's comment. The elevation of the glacier at a given bearing line is not formally part of our model. We estimate $\delta h$ by calculating $\delta l$, the change in the length of bearing lines through modelled and mapped shadows, using a trigonometric relationship, see Fig. 1d in our previous manuscript. What we understand from

the reviewer's comment is that values of $\delta l$ might correlate with their associated elevation on the glacier. In other words, as we go from lower elevations to higher elevations along the rim of shadow, the reviewer assumes that values of $\delta l$ might increase, if positively correlated, or decrease, if negatively correlated with glacier elevation. However, the shape of the shadow depends on the shape of the adjacent mountain, not the glacier, so that we expect to find a high variance of $\delta l$, even at a constant elevation of the glacier. The example of Mitre Peak (see Figure **R1A2** in the reply to Reviewer 1) underlines this concept: although Baltoro Glacier is flat (little change in elevation in the shadow covered area), bearing lines through that shadow through that shadow will have large variance.

**R2C8:** Currently, there is no consideration of elevation change uncertainty between the DEM comparisons in Section 3.3. The scanned topographic map and the SwissALTI3D DEM of Grosser Aletschgletscher for example may have substantial offsets, which are usually quantified from neighbouring stable terrain. Since the North America DEMs have gone through peer-review, they must have an associated uncertainty in the publication. For Grosser Aletschgletscher, I suggest validating that differences on stable terrain are close enough to zero and putting it in writing, or in a new figure. Without considering this source of error, I would not trust the validation from Switzerland, and therefore cannot trust the statement of similar trends on L230.

**R2A8:** We agree with the reviewer that we did not assess the uncertainties of the comparison of the topographic maps and the SwissALTI3D DEM. Unfortunately, we do not have access to the georeferenced topographic data but need to rely on manual measurements on the web-interface of swisstopo.admin.ch. However, in our revisions, we will address this point and measure points of stable terrain to exclude that measurement errors or changes in vertical datum of the maps have caused a trend which is absent in reality.

**R2C9:** Please validate the date of the ArcticDEM product you used over Gulkana Glacier. The ArcticDEM explorer only shows strips from 2011–2016 in that location, while you claim that your DEM is from 2009.

**R2A9:** We confirm that our strip is from June 16, 2009, and the file we used is: SETSM_WV01_20090616_10200100079A2600_1020010007D06000_seg1_2m_v3.0.tif

**Specific comments:**

**R2C10:** L18: Mention that the thinning pertains to the points of measurement, not the entire glacier. Otherwise, people might cite this number incorrectly.

**R2A10:** We agree that this and the previous statement could be misleading. We will revise this phrase to: **"We validated the relative differences with in situ geodetic measurements of glacier elevation change where these shadows occurred. We find that shadow-derived glacier elevation changes are consistent with independent photogrammetric and geodetic surveys in shadowed areas. Accordingly, a shadow cast on Baltoro Glacier**

**(Karakoram, Pakistan) suggests slight local increases in elevation between 1987 and 2020, while shadows on Great Aletsch Glacier (Switzerland) point to the most negative thinning rates of about 1 m per year. Our estimates of glacier elevation change are tied to the occurrence of mountain shadows, and may help complement field campaigns in regions that are difficult to access."**

**R2C11:** L36: Is 141,000 km³ for mountain glaciers or total land ice?

**R2A11:** The value is for all glaciers in the Randolph Glacier Inventory, and we used the estimate from Millan et al. (2022).

**R2C12:** L37: Change "cover" to "covers"; magnitudes are in singular.

**R2A12:** km³ (cubic kilometres) are plural, so our statement was correct.

**R2C13:** L40: Change "elevations" to "elevation"; magnitudes are in singular.

R2A13: We will change our wording accordingly.

**R2C14:** L41: Change "(ICESat)" to "(e.g. ICESat)".

**R2A14: We will change our wording accordingly.**

**R2C15:** L44: There are more large-scale long-term studies (e.g. Belart et al., 2020; Geyman et al., 2022; Mannerfelt et al., 2022).

**R2A15:** We thank the reviewer for making us aware of these studies. **We will cite them in the revised manuscript.**

**R2C16:** L48: All Landsat programme products do not have a resolution of 30 m. Only most bands of the TM, ETM+ and OLI instruments (see the Landsat program wikipedia article).

**R2A16:** We agree and will add **"a moderate spatial resolution of 30 m in the visible to shortwave infrared electromagnetic spectrum".**

**R2C17:** L49: Change "mapped in Landsat" to "mapped from Landsat"

**R2A17: We will change our wording accordingly.**

**R2C18:** L51: Change "help reveal" to "be used to estimate" or similar. Outlines themselves cannot reveal elevation change. The first reference uses area for a rough estimation of volume change, and the second uses them to crop their DEM differencing.

**R2A18: We will change our wording accordingly.**

**R2C19:** L64: The DEM abbreviation is already introduced in the abstract, and the DEM abbreviation is used in L49.

**R2A19: We will delete "Digital Elevation Models" to shorten the text.**

**R2C20:** L81: Change "long(decadal)" to "decadal"

**R2A20: We will change this statement accordingly.**

**R2C21:** L86: The line starting with "High and steep" is very short, and could be rephrased or combined with another.

**R2A21:** We strive to keep our wording simple and would therefore like to keep this sentence unchanged.

**R2C22:** L87: Specify that "Concordia" is the name of a mountain.

**R2A22:** In our manuscript, we had written that Concordia (~4,500 m a.s.l.) is the confluence of Baltoro and Godwin-Austen Glacier. **We will add this location also in Figure 2.**

**R2C23:** L89: Change "Gulkana" to "Gulkana glacier" to be consistent with L90 and to help the reader. I know that the word "glacier" occurs in the beginning of the sentence, but the wording is still not obvious to me.

**R2A23: We will change our wording accordingly.**

**R2C24:** L105: Did you validate the georeferencing? Sometimes they can be off if not enough GCPs exist. Please just mention that this was (hopefully) not a problem.

**R2A24:** We could not find any inconsistencies and will add to our revised manuscript: **"L1TP images offer high radiometric and geodetic accuracy by using ground control points and correcting for topographic displacement using regional DEMs (https://www.usgs.gov/landsat-missions/landsat-levels-processing#L1TP). We could not find any notable offsets between successive images."**

**R2C25:** L109: Which ArcticDEM product was used? The mosaic or an individual strip? If an individual strip, which id?

**R2A25: We used individual strips, and will add the individual IDs to Table A2.**

**R2C26:** L112: Do you mean the swissALT3D DEM 2019 version? The acquisition year over Grosser Aletschgletscher is 2017–2018 as is correctly mentioned later.

**R2A26:** We are unsure about the reviewer's comment.

**R2C27:** L122: It would be nice to know a bit more about why the green band was chosen. For example, did it have the highest difference between shaded and unshaded locations? It is mentioned that "shadows appear dark" in the green band, but not "darker" than the other bands.

**R2A27:** In our revised manuscript, we will add: **"Snow, firn, and ice have minimal absorption in the blue-green range, whereas red and infrared light is strongly absorbed on these surfaces. This trait enhances contrast at the interface of glaciated surfaces and shaded, colder areas with increasing wavelength. Incoming and reflected electromagnetic wavelength in the green band is also less affected by the Rayleigh scattering in the atmosphere compared to the blue band that has a shorter wavelength. The green band therefore offers good compromise between contrast and surface reflectance measured at the sensor"**.

**R2C28:** L125: Does the algorithm in SAGA account for the Earth's curvature? The effect is in the order of 1 m in elevation per 3 km of horizontal distance, so this would matter somewhat at these scales. Please just mention if it is already accounted for, or mention that future improvements can be made by accounting for it.

**R2A28:** Olaf Conrad, the core developer of SAGA, emailed us that the algorithm 'Analytical hillshading' does not account for Earth's curvature. Yet we expect, if at all, a minor role of Earth's curvature in our results. The longest bearing line that we modelled had a length $d$ of 308.73 m (Gulkana 2019). We can calculate the deviation $a$ of a plane from the spherical surface (radius $r$) with increasing distance $d$ from the point of contact (plane to sphere) using the theorem of Pythagoras: $a = \sqrt{d^2 + r^2} - r$. For $d$ = 0.30873 km and $r$ = 6371 km, i.e. the mean radius of the Earth, we obtain a deviation $a$ = 0.0007480318 km (i.e. 7.4 mm) from Earth's surface. In our analysis, however, we use only the much shorter segment of the bearing line that extends beyond the modelled shadow, and therefore obtain even smaller values for $a$.

**R2C29:** L128: Specify that it's RGI version 6.

**R2A29: We will revise this statement to "Randolph Glacier Inventory (RGI) V6.0".**

**R2C30:** L187: Are you entirely sure that the maps represent these exact years? Swiss maps are very often asynchronously updated over time, and can have large (tens of metres) differences to what one would expect (c.f. Fig. 10 in Mannerfelt et al., 2022).

**R2A30:** We used the years at which the topographic maps were available first in the online interface of swisstopo (link). As stated above, we will aim to detect possible offsets between SwissALTI3d data and the maps on stable terrain. The paper by Mannerfelt et al. (2022) provides useful backgrounds and details for this analysis.

**R2C31:** L207: The swissALTI3D DEM is a mix of Lidar at low elevation and aerial photogrammetry at high elevation (see the technical information on swisstopo's website). At Grosser Aletschgletscher, it is therefore most likely based on photogrammetry.

**R2A31:** Yes, you are right. Information by swisstopo (link) shows that the area of the Great Aletsch glacier was last updated in 2019. The update was based on photogrammetric aerial imagery from 2016-2018. We will update the description of the data in a revised version of the manuscript.

**R2C33:** L211: Repeat what the line spacing is. 45 lines is not informative unless the spacing is mentioned.

**R2A33:** In line 129, we had written that we use a regular spacing of 30 m between the bearing lines, and will add this information here again.

**R2C34:** L231: The topographic maps may have a high resolution, but it is far from guaranteed to have a high accurancy and precision. I would rephrase this from "high-resolution DEMs" to "reference DEM comparison".

**R2A34: We will change this statement accordingly.**

**R2C35:** L244: Is the substantial variance found before or after considering the elevation dependent elevation change that is observed on almost every glacier in the world? If they are at relatively similar elevation, then say that: "... in spite of their similar elevation ...".

**R2A35:** We will change our wording to: **"The elevation changes obtained from bearing lines have substantial variance in a given year despite covering a small range in elevation along the glacier**

**R2C36:** L252: Where is the date of the GLO-90 DEM information from? On the associated OpenTopography website it says 2011–2015

**R2A36:** Yes, you are right. The data were acquired through the TanDEM-X mission between 2011 and 2015. We will correct this mistake in a revised version of the manuscript.

**R2C37:** L277: Right now, you only show a difference in the horizontal variability that is associated with different DEMs, not the effect on calculated elevation change. My Figure 7 suggestion below would solve this problem.

**R2A37:** We would like to refer the reviewer to our reply **R2A46**.

**R2C38:** L294: The effects of SAR penetration would lead to a potential negative bias (longer shadows), no? But since this bias will be consistent between years, one would just have to subtract the shadow-derived elevation change at the year of acquisition of the DEM. So this is arguably a very simple fix. Could this be put in words, assuming you agree with me?

**R2A38:** We agree, and will add: **"Our approach does not account for any offsets in elevation due to the SAR signal penetrating into snow. Yet this offset can be treated as constant when drawing bearing lines through shadows, given that the input DEM (SRTM) remains unchanged in our analysis."**

**R2C39:** L302: "Precision" (spread) AND "accuracy" (bias). There seems to be both consistent and inconsistent errors in the shadow-maps derived from poor DEMs.

**R2A39: We will add "accuracy" accordingly.**

**R2C40:** L309: Please elaborate on why you think winter months are better suited for this method. Lower solar angles, I presume? What about the contrast between shaded and unshaded terrain?

**R2A40:** Yes, mainly because of sun angles (and calculation is not influenced by the climate change related shift of starting point of melting season as well as summer heat correlated differences in melting rates (but by snowfall). On top of that we thereby wanted to make sure that the selected season of the year of every Landsat scene acquisition matches the time when SRTM was acquired.

**R2C41:** L330: Steep topography is arguably not a requirement; just stable topography. Indeed, steep slopes increase the potential time of day at which shadows can be created, so it works better with steeper topography. But the approach still works in shallow topography; just at fewer hours of the day! On e.g. ice caps, there is no stable topography to cast shadows, so the approach fails. So steep topography and high latitudes are preferable, but stable topography is the only theoretical requirement.

**R2A41:** We agree and will rewrite this statement to: **"Our method can be applied globally, but is restricted to those glaciers that are surrounded by stable topography. Ideal environments for our approach are glaciers close to steep topography in high latitudes, producing cast shadows long enough to infer differences in bearing lines".**

**R2C42:** Figure 1: I just want to add that I think this is a great figure that explains the concept simply and artfully! Great job!

**R2A42:** We appreciate the kind words of the reviewer.

**R2C43:** Figure 2: The Randolph Glacier Inventory is mentioned without abbreviating it, contrasting it to Figure 3.

**R2A43: We will introduce the abbreviation in Fig. 2 and in the caption of Fig. 3, will delete the full name and use the abbreviation only.**

**R2C44:** Figure 3: The RGI abbreviation is introduced multiple times. Only RGI is necessary here.

**R2A44: We will reduce the redundant abbreviations.**

**R2C45:** Figure 5: It is unclear where the grey bubbles come from. Are these from other satellite images than Landsat? Please clarify this in the text.

**R2A45:** All bubbles are from other sources than Landsat. We will change the caption to: **"Grey bubbles are historical data (Siegfriedkarte) obtained before the Landsat period".**

**R2C46:** Figure 7: I recommend to add another axis label on top where you show calculated height change assuming the solar parameters you decided. This would solve my issue in L277 that you have actually not shown the variable DEM quality effect on elevation change.

**R2A46:** This is a very good idea. We will add a second x-axis to map horizontal values to vertical offsets given the chosen azimuth and sun elevation.

**References:**

Belart JMC, Magnússon E, Berthier E, Gunnlaugsson AT, Pálsson F, Aðalgeirsdóttir G, Jóhannesson T, Thorsteinsson T, Björnsson H. 2020. Mass Balance of 14 Icelandic Glaciers, 1945–2017: Spatial Variations and Links With Climate. Frontiers in Earth Science 8:163. doi: 10.3389/feart.2020.00163.

Geyman EC, van Pelt WJJ, Maloof AC, Faste Aas H, Kohler J. 2022. Historical glacier change on Svalbard predicts doubling of mass loss by 2100. Nature 601:374–395. doi:10.1038/s41586-021-04314-4.

Mannerfelt ES, Dehecq A, Hugonnet R, Hodel E, Huss M, Bauder A, Farinotti D. 2022. Halving of Swiss glacier volume since 1931 observed from terrestrial image photogrammetry. The Cryosphere Discussions :1–32doi:10.5194/tc-2022-14.

**REFERENCES**

Hugonnet, Romain; McNabb, Robert; Berthier, Etienne; Menounos, Brian; Nuth, Christopher; Girod, Luc et al. (2021): Accelerated global glacier mass loss in the early twenty-first century. In *Nature* 592 (7856), pp. 726–731. DOI: 10.1038/s41586-021-03436-z.

Millan, R.; Mouginot, J.; Rabatel, A.; Morlighem, M. (2022): Ice velocity and thickness of the world's glaciers. In *Nature geoscience* 15 (2), pp. 124–129. DOI: 10.1038/s41561-021-00885-z.

U.S. Geological Survey (2023): Landsat Levels of Processing. Available online at https://www.usgs.gov/landsat-missions/landsat-levels-processing, updated on 2/13/2023, checked on 2/13/2023.

---

## Author Comment (AC3)

**Comment by Niccolò Dematteis**

We thank Niccolò Dematteis for making us aware of this recent publication. In this reply, we highlight the comments from Dr Dematteis in orange and our responses in black font.

**R3C1:** Dear authors, I read with interest your work, as I am involved in similar research. I would like to inform you about a very recent publication of ours (https://ieeexplore.ieee.org/document/9997554). Our work focuses on the same topic that is treated in your manuscript, starting from the same concept but solving the issue differently. As evident, in the very last years, different research teams worked on this topic independently, thus proving the interest in this theme among the scientific glaciological community.

**R3A1:** We thank the reviewer for this positive appraisal and recognition of our work. Unfortunately, this interesting paper was not yet available when we submitted our manuscript, but we will include it a revised manuscript.

**R3C2:** I have a curiosity about your work: are the Landsat images orthorectified on the same DEM? If not, can a certain bias verify if the case of the adopted DEMs are acquired at large temporal distances (since the glacier elevation where the satellite images are projected has likely varied)?

**R3A2:** We used orthorectified Landsat imagery which is processed using different sources of elevation data including SRTM, NED, and other DEMs. We used the SRTM as well as other DEMs that are not among those used for the terrain correction. It is possible that long periods between image acquisition and DEM acquisition might introduce some bias. Visual analysis of imagery of glacial surfaces did not reveal any such bias.

---

## Referee Report (RR1)

**General comments**

It is evident that the authors have put much work into improving the readability and overall quality of this manuscript. I can gladly say that I think they succeeded, and I only have small rephrasing suggestions! I recommend this for acceptance with minor technical corrections for publication.

**Line-specific comments**

Below, the line numbers represent those in the revised manuscript.

**L45**: All these new references are from work based on aerial and terrestrial photographs. I suggest to move the reference before the "Corona and Hexagon" satellite names, or to rephrase the sentence to fit the references better.

**L120**: It would be nice with a second reference to your appendix table with the ArcticDEM ID here.

**L126**: This is the first time SRTM-3 is mentioned. Could you clarify in a parenthesis what the difference is to SRTM-1? (It's stated further down, but the reader is introduced to the abbreviation here)

**L135-141**: Great explanation!

**L145**: I believe the second "shadow" in this sentence should be in plural.

**L234-235**: "[…] shows a sinusoidal up and down." sounds like a word is missing. Perhaps "[…] shows a sinusoidal variation up and down throughout the seasons."?

**L265**: Please change "[…] 10 times that of […]" to "[…] 10 times more negative than […]" or similar to lower ambiguity about this statement (it's unclear in which direction it is 10 times different).

**L418**: If you wrote "less negative" instead of "lower" the sentence would read better in my opinion. -1.42 is lower than -1.08, but the magnitude is the opposite (as I presume you allude to). I find it less confusing to write "less negative".

**L488**: Romain is indeed a "hepful" person! (Little typo in "help")

---

## Author Response (AR2)

University of Potsdam **|** Institute of Environmental Science and Geography

Karl-Liebknecht-Str. 24-25 | 14476 Potsdam-Golm | GERMANY

[Figure]

Etienne Berthier

Editor at *The Cryosphere*

**via email**

**Faculty of Science**
**Institute of Environmental Science and Geography**

**Monika Pfau**
*email:*    monika.pfau@uni-potsdam.de
*Date:*    2023-07-19

Dear Dr. Berthier,
Dear Editors,

thank you for giving us the opportunity to further revise our manuscript tc-2022-194 entitled **"Cast shadows reveal changes in glacier surface elevation"**. We addressed all comments and questions from the editor and the two reviewers in a point-by-point reply letter, and are happy to present a revised version of our manuscript.
In our reply letter, comments from the editor (EC) and reviewers (RC1 and RC2) are blue, and our answers are in black font. References to specific lines refer to the revised manuscript.

We look forward to hearing from your decision in due course.

Yours sincerely,

Monika Pfau
with co-authors Georg Veh and Wolfgang Schwanghart
* * *
**From the editorial board, we received the following comments:**

**EC1:** Both reviewers recognized the large amount of work made to improve the manuscript. One is now fully satisfied with the revised text whereas the other reviewer still challenge the use of View Finder Panorama as the best DEM for the Baltoro case study. A sensitivity analysis, describing the difference of dh on Baltoro when using SRTM1 instead of VFP (as nicely down for Aletsch) should help to reconcile both views and strengthen further your study. (I wondered whether you and the reviewer used the same version of SRTM1. If you want me to ask his version of SRTM1 do not hesitate to contact me directly by email.)

**EC1A1:** We acknowledge that the use of the DEM from Viewfinder Panorama (VFP) has attracted considerable criticism from one of the reviewers, fuelling a debate that we had not anticipated. Indeed, it is difficult to show whether the void-filled SRTM or the VFP better represents Mitre Peak near Baltoro Glacier because the interpolation method remains unknown in both cases. However, the reviewer's assessment was based solely on a visual assessment of a hillshade, and the conclusions drawn remain subjective and conjectural. For example, the reviewer argued that *"in view of this, it seems obvious that even if underlying filling data are unknown, there SRTM 1" can be seen as a better candidate for this study which weakens the rebuttal. The assessment of peak elevation is clearly affected by large uncertainty and unconvincing I believe to conclude."* This assessment is free from any quantitative support that SRTM-1 has a better interpolation algorithm or smaller uncertainties than VFP. We had clearly pointed out in the manuscript and in the reply letter that there is no independent high-resolution DEM for this peak to quantify possible offsets between VFP or SRTM from a reference surface. The 8-m HMA-DEM also features extensive voids at Mitre Peak.

The difference between VFP and SRTM suggests that the peak in VFP is 154 m higher than in SRTM. A profile drawn across Mitre Peak in both DEMs shows that the elevation in steep

[Figure]

*Figure 1: Comparison of elevation from two DEMs at Mitre Peak, adjacent to Baltoro Glacier, Pakistan. Upper panels show color-coded elevation values at Mitre Peak in SRTM and VFP DEM draped over a hillshade. Lower panels show that the difference between the two DEMs can be several hundreds of meters higher. The black line in the lower left panel shows the location of the transect in the lower right panel.*

topography is higher in VFP, while elevations on the flat glacier surface are identical (Figure 1). Geoid heights (differences between the geoid and ellipsoidal elevations) are ~22 m in this region. Thus, we exclude the possibility that both DEMs have different vertical datums. We thus conclude that only the shape and height of Mitre Peak can cause differences in modelled shadows.

In Figure 2, we compare the shadows from SRTM, acquired in 2000, and VFP. Accordingly,

[Figure]

*Figure 2: Hillshades including modelled shadows of the Mitre Peak adjacent to Baltoro Glacier, Pakistan. We used the SRTMGL1 and replaced the Mitre Peak with different DEMs. Shadows were calculated with an azimuth of 151.9° and a sun elevation angle of 29.5°. These values refer to the sun position during the acquisition time (Jan 24, 2000) of the Landsat image from which shadows were mapped (red outline). Visual comparison shows that the SRTM+VFP creates the best match between modelled and actual cast shadows, whereas there are pronounced offsets between actual shadows and those derived from other DEMs.*

the modelled shadow from VFP is slightly longer than the mapped shadow from Landsat imagery, while that from SRTM is too short (Figure 2), consistent with the differences in peak elevations. VFP is based on SRTM, but our comparison clearly shows that the VFP of the Mitre Peak has not been calculated by simple void filling of SRTM-3 data, but we rather speculate that other data were taken into account (e.g. Russian topographic maps as in the Hispar Muztagh, Karakorum, http://viewfinderpanoramas.org/elevmisquotes.html#asia).

We conclude that both DEMs are suitable for approximating the geometry of the shadow at Mitre Peak, but both have their limitations. This is not surprising given the steep topography of Mitre Peak, whose gradient is difficult to represent in DEMs with 30 or 90 m cell resolution (Figure 3).

[Figure]

*Figure 3: Steep topography at Mitre Peak. Photograph by By Anne Dirkse (www.annedirkse.com) - Own work, CC BY-SA 4.0, https://commons.wikimedia.org/w/index.php?curid=34911367*

**EC2:** I also agree with his point that the comparison to Hugonnet et al. rate of elevation change should be performed only on the common pixels, hence at the edge of the cast shadow and not over the entire shadowed glacier area.

**EC2A2:** We agree and asked Romain Hugonnet to provide the same summary statistics only for the pixels covering the area between the edges of the smallest and the largest shadow mapped in satellite images. We opted for an area rather than a single line along the edge of the shadow, given that the shadow extents in satellite images varies in our study period

[Figure]

**Figure 3: Area between the smallest/ shortest (2014) and the largest/ longest (1991) shadow cast from Dreieckhorn onto Great Aletsch Glacier (Switzerland).** *Lines within the yellow area are all other 21 shadow outlines obtained between 1987 and 2019. Image in the background is a Bing Aerial basemap available through QGIS.*

(Figure 4). The example of the Great Aletsch Glacier shows how small the spacing of the shadow edges is in our study period, so that it is difficult to decide on exactly one of them. Choosing the area between the largest/ longest and the smallest/ shortest shadow better reflects the variance in glacier elevation change through time and provides a good compromise for comparing our data with those of Hugonnet et al. (2021).

Nevertheless, we did not find any substantial differences between the earlier and our updated analysis after running the models with the modified datasets (except that we had confused the trend at Gulkana West with that of Gulkana East in Figure 4). Thus, the revised analysis confirms the generally good agreement between our data and those of Hugonnet et al. (2021). Drivers for some of the diverging trends, as observed at Gulkana East, are now discussed in more detail in the discussion.

**EC3:** L235. It is unconventional to acknowledge a colleague in the main text. This belongs to the acknowledgement section (where you need to correct "help"). Hence you could reformulate L235 and rather cite a personal communication for a peculiar data extraction.

**EC3A3:** We reformulated this sentence to (L236-238): "*We used time series of glacier elevation change extracted along simplified outlines of glacier shadows (Fig. 2), provided as summary statistics on mean glacier elevation change between 2000 and 2019 by Romain Hugonnet (pers. comm., 2023) (Fig. 6)*".

**EC4:** Figure 5. The legend needs to explain which results are from DEMs and which are from cast shadows.

**EC4A4:** We added a legend to this figure accordingly.
* * *
**From Reviewer #1 (R1), we received the following comments:**

**R1C1:** L45: All these new references are from work based on aerial and terrestrial photographs. I suggest to move the reference before the "Corona and Hexagon" satellite names, or to rephrase the sentence to fit the references better.

R1A1: We agree and rewrote the sentence to read as (L43-45): "*These appraisals are largely constrained to the past two decades (Belart et al. 2020; Geyman et al. 2022; Mannerfelt et al. 2022), with few exceptions such as Corona and Hexagon missions, which provided one-time stereo image pairs between the 1960s and 1970s (Lovell et al. 2018; Dehecq et al. 2020).*"

**R1C2:** L120: It would be nice with a second reference to your appendix table with the ArcticDEM ID here.

**R1A2:** We added a reference to Table A2 accordingly.

**R1C3:** L126: This is the first time SRTM-3 is mentioned. Could you clarify in a parenthesis what the difference is to SRTM-1? (It's stated further down, but the reader is introduced to the abbreviation here).

**R1A3:** In our revised manuscript, we have deleted the paragraph where this sentence was located. We have also deleted the entry regarding SRTM-3 in Table A2.

**R1C4:** L145: I believe the second "shadow" in this sentence should be in plural.

**R1A4:** We used the plural instead.

**R1C5:** L234-235: "[...] shows a sinusoidal up and down." sounds like a word is missing. Perhaps "[...] shows a sinusoidal variation up and down throughout the seasons."?

**R1A5:** We changed this phrase so that it reads (L235-236): "… time series of glacier elevations show seasonal variations".

**R1C6:** L265: Please change "[...] 10 times that of [...]" to "[...] 10 times more negative than [...]" or similar to lower ambiguity about this statement (it's unclear in which direction it is 10 times different).

**R1A6:** We changed this statement accordingly.

**R1C7:** L418: If you wrote "less negative" instead of "lower" the sentence would read better in my opinion. -1.42 is lower than -1.08, but the magnitude is the opposite (as I presume you allude to). I find it less confusing to write "less negative".

**R1A7:** We agree and changed the wording accordingly.

**R1C8:** L488: Romain is indeed a "hepful" person! (Little typo in "help")

**R1A8:** We changed "hep" to "help" accordingly.
* * *
**From Reviewer #2 (R2), we received the following comments:**

**R2C1:** R1C2/R1A2: With respect to my comment on the use of VFP vs SRTM 1'', I agree that the SRTM 1'' has void filled and is not more explicit than VFP about the source of data in this area. It is also true that both mountains casting shadows over Baltoro are those affected by voids. The DoD in Figure R1C1 reveals very well the extent of those voids, while also demonstrating that data used to fill either VFP or SRTM 1'' are very different. That being said,

the sole appearance of hillshade DEMs from VFP or SRTM 1'' arguably suggests that SRTM 1'' conveys substantially better resolution and details which I believe should justify its use over VFP regardless. It also suggests that the voids in SRTM 1'' are not simply interpolated in this instance as the hillshade reveals topographic details that an interpolation may hardly achieve. Following on this, the authors now cite Fig. 6 in Liu et al. 2019 to support higher accuracy of VFP in steep terrain. For some reason, the reference provided in the rebuttal is wrong (namely, Liu, Xiaodong; He, Pengcheng; Chen, Weizhu; Gao, Jianfeng (2019): Improving Multi-Task Deep Neural Networks via Knowledge Distillation for Natural Language Understanding. Available online at https://arxiv.org/pdf/1904.09482.). The reference provided in the revised paper is correct. Nevertheless, I am confused by this argument. It is true that Fig 6 in Liu et al. 2019 suggests MAE of SRTM 3'' is greater than VFP, yet MAE of SRTM 1'' is reported as substantially better regardless of slope. I am left puzzled by this result not making the author reconsider the use of VFP, since they construe it as invalidating the use of SRTM 3''.

As the authors explain themselves, DEMs of higher resolution might better preserve the distinct shape of the mountains. In view of this, it seems obvious that even if underlying filling data are unknown, there SRTM 1'' can be seen as a better candidate for this study which weakens the rebuttal. The assessment of peak elevation is clearly affected by large uncertainty and unconvincing I believe to conclude. That being said, I find that SRTM 1'' resolves Mitre Peak with an elevation of 5994m above EGM96 which is not as different from the estimated elevation of the peak as the author suggest. In fact, it is unclear why the authors report a height of only 5904m from SRTM 1'' in Fig R1A2 which undermines again the rebuttal of this comment.

Overall, I remain unconvinced by this answer and still believe the relevance of using VFP is weak and unfounded.

**R2A1:** We note that the discussion about the shape and height of Mitre Peak has taken on an intensity that we had not anticipated. First of all, we agree with the referee that SRTM void filling is not simply interpolation. In fact, version 3.0 has been filled with ASTER GDEM version 2.0. This, however, may not necessarily improve the SRTM DEM in terms of accuracy as the ASTER GDEM has severe issues in mountainous terrain (see http://www.viewfinderpanoramas.org/reviews.html#aster). Second, we cannot reconstruct the elevation of 5994 m of Mitre Peak in SRTM 1", which the reviewer had mentioned. We obtained the DEM from opentopography (www.opentopography.org) and the DEM has the vertical datum EGM96. In our reply to the editor (EC1A1 and Figure 1), we showed that different vertical datums cannot cause the possible offset between the SRTM and VFP because glacier elevations in both DEMs are the same. In our comparison of peak elevations, we use the unprojected DEMs so that we can also exclude that gray-value interpolation during reprojection causes lowering of the elevations. De Ferranti (the producer of the VFP) replaced missing values with data obtained from topographic maps. The higher values of the Mitre Peak in the VFP data suggests, that this data is eventually more accurate than any of the DEMs.

The main manuscript now contains a section in which we explain the comparison of shadows retrieved from the different DEMs as well as Figure 2 shown above (Figure 10 in the revised manuscript).

**R2C2:** I am unconvinced that the comparison with Hugonnet et al (2021) should account for all pixels inside the shadow. Shouldn't it rather be only those pixels mapped along the edge of the shadow instead?

**R2A2:** We agree and asked Romain Hugonnet for the same summary statistics only for the pixels at the edges between the largest and smallest mapped shadow. We assume that these shadows are the endmembers of the variance in shadows derived from Landsat images in our study period. Re-running our analysis using the new data did not change our findings (Figure 5); however, the trends at Great Aletsch Glacier agree better between the two methods. With the exception of one year on the Great Aletsch and Gulkana East glaciers, the Gaussian process regression models of Hugonnet et al. (2021) overlap with our data (interquartile ranges of the boxes), indicating good agreement between the two methods.

[Figure]

***Figure 4: Comparison of the original Figure 6 (top panels) and revised Figure 6 (lower panels).*** *The trends calculated from simplified shadow outlines (top) largely agree with those obtained from the area between the shortest and longest shadow on each glacier. Note that we had confused Gulkana East with Gulkana West in our previous submission. We also improved the legend.*

**R2C3:** As per my comment above, I don't understand why all pixels in shadow are used since the method can only capture changes on the edge of the shadow.

**R2A3:** Please see the answer **R2A2** above.

**R2C4:** Further to that, the response provided in R1A4 "suggest slight local increases" is now "suggests no change" in the revised manuscript. This is a rather significant change in conclusion which further underpins what I still believe is a limitation of the method, and at least deserves more explanation as to why the inference has changed given I understand the authors did not change the DEM. In effect, it appears the data for Baltoro in revised Figure 4 have been modified compared to the original Figure 4 which led to a different statistical model. This requires clarification.

**R2A4:** We thank the reviewer for bringing this change in our last response letter to attention. Following our initial analysis, we had identified a few bearing lines that had lengths far outside the expected frequency distribution, and excluded them from the analysis. In addition, some bearing lines were wrongly classified as lines connecting modelled shadows with mapped shadows whereas they were actually connecting one of those shadows with the outline of the Randolph Glacier Inventory outside the area of the second shadow. We had made those adjustments to the dataset for all considered years. To this end, these changes led to different results compared our very first manuscript version.

**R2C5:** Finally, the authors dispute my statement that Hugonnet et al. (2021) shows the trend at Baltoro is "unambiguously negative". They argue that on a pixel basis, the uncertainty exceeds the elevation change. Although that latter is correct, it cannot be construed into defeating my argument since Figure R1C4(a) reveals the wide-spread negative value. Suggesting that uncertainty on a pixel basis applied to the overall budget, or even a subset of pixels, is erroneous. It corresponds to confusing standard deviation applying to a single measurement compared to standard error applying to the average. For this argument to be valid, it would imply that Huggonnet et al (2021) data are biased, which is not to be the case by design. Suggesting that the authors data are more accurate on a local scale can only be supported by a rigorous analysis of the distribution of values from pixels along the shadow with the standard error being considered in view of the sample size.

**R2A5:** We thank the reviewer for clarifying their former statement. We believe that there is a misunderstanding in the interpretation of the phrase "unambiguously negative". We interpreted this phrase only such that every measurement (cell value) of glacier elevation change, i.e. the mean **and** the error associated with that mean estimate), in the data from Hugonnet et al. (2021) needs to be smaller than zero to be "unambiguously negative". In showing the measurements for every raster cell, we showed that the trend in mean elevation change in any cell is both positive and negative, assuming one standard deviation error, regardless whether this cell was extracted at the edge of the shadow or not. In addition, some individual pixels even had positive mean values of glacier elevation change. We did not talk about the "overall budget" of the observed changes from Hugonnet et al. (2021), nor did we conclude that their

methods or results are "biased". Indeed, our manuscript acknowledges their elaborate method and the high value of their dataset.

**R2C6:** Thank you for addressing separately both areas of Gulkana. Nonetheless, I am sceptical again that the proposed method suggests a rate of thinning seven times (not 10 times as written in Section 4) faster in the West (Ogive mountain) compared to East (Icefall Peak), namely -1.58 compared to -0.22 m a-1. Again, Figure R1C4(b) provides key insight to compare glacier change in both areas. Although the authors signal this discrepancy, they unfortunately come short of providing a convincing explanation or stressing what I believe could be exemplifying the limitation on the method.

**R2A6:** We are unsure where these values of glacier elevation change at Gulkana Glacier come from as we could not find those in our revised text or figures. In any case, we agree that there are differences, both in the method and in the results between our appraisal and that of Hugonnet et al. (2021), let alone any other appraisal using remote sensing or field work. In our manuscript, we had written: *"One reason for the discrepancy between the two datasets may be the rigorous filtering of outliers in the dataset of Hugonnet et al. (2021), whereas our method maintains the elevation changes of all bearing lines, regardless of their distances from the mean or median"*. In the discussion, we now point at the advantage of using more informed priors from other glaciers or studies to reconcile the posterior trends, even if the physical or methodological drivers of the underlying trends remain unknown (L425-432): *"In any case, our Bayesian framework objectively propagates these errors and uncertainties. One promising avenue for future research is to use more informed priors based on previous research on glacier elevation change (Hugonnet et al. 2021). Narrower and stronger priors may reduce the width of our posterior trends on glacier elevation changes that we currently observe at Sperry Glacier, for example (Fig. 4). They might also offer a better compromise to balance some of the differences within our data (e.g. between Gulkana East and West), and also between our data and data from previous research. One of these examples may be the outstanding trend at Gulkana West (Fig. 6), where the physical causes and methodological differences between our appraisals and that of Hugonnet et al. (2021) remain to be determined."*

---

## Author Response (AR3)

University of Potsdam **|** Institute of Environmental Science and Geography

Karl-Liebknecht-Str. 24-25 | 14476 Potsdam-Golm | GERMANY

[Figure]

Etienne Berthier

Editor at *The Cryosphere*

**via email**

**Faculty of Science**
**Institute of Environmental Science and Geography**

**Monika Pfau**
*email:* monika.pfau@uni-potsdam.de
*Date:* 2023-07-25

Dear Dr. Berthier,

thank you for accepting our paper tc-2022-194 entitled **"Cast shadows reveal changes in glacier surface elevation"** and for your extra effort.
We have made the final corrections you suggested and look forward to seeing the manuscript published.

Yours sincerely,

Monika Pfau
with co-authors Georg Veh and Wolfgang Schwanghart